

# Towards monitoring localized $CO_2$ emissions from space: co-located regional $CO_2$ and $NO_2$ enhancements observed by the OCO-2 and S5P satellites

Maximilian Reuter[1], Michael Buchwitz[1], Oliver Schneising[1], Sven Krautwurst[1], Christopher W. O'Dell[2], Andreas Richter[1], Heinrich Bovensmann[1], and John P. Burrows[1]

[1]Institute of Environmental Physics, University of Bremen, Germany
[2]Colorado State University, Fort Collins, CO, USA

**Correspondence:** Maximilian Reuter (mail@maxreuter.org)

**Abstract.** Despite its key role for climate change, large uncertainties persist in our knowledge of the anthropogenic emissions of carbon dioxide ($CO_2$) and no global observing system exists allowing to monitor emissions from localized $CO_2$ sources with sufficient accuracy. The Orbiting Carbon Observatory-2 (OCO-2) satellite can retrieve the column-average dry-air mole fractions of $CO_2$ ($XCO_2$). However, regional column-average enhancements of individual point sources are usually small com-

5 pared to the background concentration and its natural variability. This makes the unambiguous identification and quantification of anthropogenic emission plume signals challenging. $NO_2$ is co-emitted with $CO_2$ when fossil fuels are combusted at high temperatures. It has a short lifetime of the order of hours so that $NO_2$ columns often exceed background levels by orders of magnitude near sources making it a suitable tracer of recently emitted $CO_2$. Based on six case studies (Moscow, Russia; Lipetsk, Russia; Baghdad, Iraq; Medupi and Matimba power plants, South Africa; Australian wildfires; and Nanjing, China),

we demonstrate the usefulness of simultaneous satellite observations of $NO_2$ and the column-average dry-air mole fraction of $CO_2$ ($XCO_2$). For this purpose, we analyze co-located regional enhancements of $XCO_2$ observed by OCO-2 and $NO_2$ observed by the Sentinel-5 Precursor (S5P) satellite and estimate the $CO_2$ plume's cross-sectional fluxes. We take advantage of the nearly simultaneous $NO_2$ measurements with S5P's wide swath by identifying the source of the observed $XCO_2$ enhancements, excluding interference with remote upwind sources, allowing to adjust the wind direction, and by constraining the

shape of the $CO_2$ plumes. We compare the inferred cross-sectional fluxes with the Emissions Database for Global Atmospheric Research (EDGAR), the Open-Data Inventory for Anthropogenic Carbon dioxide (ODIAC), and, in the case of the Australian wildfires, with the Global Fire Emissions Database (GFED). The inferred cross-sectional fluxes range from $32\,MtCO_2/a$ to $158\,MtCO_2/a$ with uncertainties ($1\sigma$) between 23% and 72%. For the majority of analyzed emission sources, the estimated cross-sectional fluxes agree within their uncertainty with either EDGAR or ODIAC or lie in between them. We assess the

contribution of multiple sources of uncertainty and find that the dominating contributions are related to the computation of the effective wind speed normal to the plume's cross-section. The planned European Copernicus anthropogenic $CO_2$ monitoring mission (CO2M) will not only provide precise measurements with high spatial resolution but also imaging capabilities with a wider swath of simultaneous $XCO_2$ and $NO_2$ observations. Such a mission, in particular as a constellation of satellites, will deliver $CO_2$ emission estimates from localized sources at an unprecedented frequency and level of accuracy.

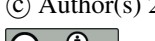



## 1 Introduction

Carbon dioxide ($CO_2$) is the most important anthropogenic greenhouse gas and driver for climate change. By September 2018, 195 member states of the UNFCCC (United Nations Framework Convention on Climate Change) have signed the Paris agreement with the long-term goal to keep the increase in global average temperatures relative to pre-industrial levels well below

2°C. Actions need to be taken to halve anthropogenic greenhouse gas emissions (including $CO_2$) each decade after reaching a maximum in 2020 (Rockström et al., 2017). However, there are still large uncertainties in the anthropogenic emissions and no global observing system exists allowing to monitor country emissions and their changes with sufficient accuracy (e.g., Ciais et al., 2014; Pinty et al., 2017).

 $CO_2$ is long-lived and well-mixed in the atmosphere and its largest gross fluxes are of natural origin (photosynthesis and

respiration). As a result, regional (column-average) enhancements of individual anthropogenic point sources are usually small compared with the background concentration and its natural variability (Bovensmann et al., 2010). This makes the identification of anthropogenic plume signals with past (SCIAMACHY (SCanning Imaging Absorption SpectroMeter for Atmospheric CHartographY, Burrows et al., 1995; Bovensmann et al., 1999)) and current (GOSAT (Greenhouse Gases Observing Satellite, Kuze et al., 2009), OCO-2 (Orbiting Carbon Observatory-2, Crisp et al., 2004)) satellite sensors difficult and the quantification

of anthropogenic emissions a challenging task. Usually, it requires knowledge of the source position and assumptions on plume formation (e.g., Nassar et al., 2017; Heymann et al., 2017; Krings et al., 2011, 2018) or statistical approaches applied on larger areas (e.g., Kort et al., 2012; Schneising et al., 2013; Buchwitz et al., 2017).

 Reuter et al. (2014) followed an alternative approach to identify anthropogenic regional $CO_2$ enhancements by analyzing simultaneous satellite observations of tropospheric nitrogen dioxide ($NO_2$) vertical columns and column-average dry-air mole

fractions of $CO_2$ ($XCO_2$). Nitrogen monoxide (NO) is formed and emitted to the atmosphere when fossil fuels are combusted at high temperatures. In the atmosphere, it reacts rapidly with ozone ($O_3$) and at a much slower rate via a termolecular reaction with oxygen ($O_2$) to form $NO_2$. The tropospheric daytime concentrations of $NO_2$ are coupled with the concentrations of NO and $O_3$ by the Leighton photo-stationary state. $NO_2$ has a short lifetime of the order of hours so that its vertical column densities often exceed background levels by orders of magnitude near sources (Richter et al., 2005) making it a suitable tracer of recently

emitted $CO_2$.

 In contrast to SCIAMACHY used by Reuter et al. (2014), OCO-2 has no $NO_2$ sensor aboard. However, with the launch of the S5P satellite (Sentinel-5 Precursor, Veefkind et al., 2012) in 2017, $NO_2$ observations with unprecedented spatial resolution and global daily coverage became available. Here we use this data to identify OCO-2 $XCO_2$ enhancements which can be attributed to localized small scale emissions for which we estimate the plume's cross-sectional $CO_2$ fluxes.

In the next section, we describe the used OCO-2 $XCO_2$ and S5P $NO_2$ data sets and the developed co-location method. In section 2, we describe the used plume detection and scenario selection method as well as the cross-sectional flux estimation method. The results of our case study analyses are presented and discussed in section 3 and 4, respectively.



## 2 Data sets and methods

### 2.1 XCO$_2$

The Orbiting Carbon Observatory-2 (OCO-2, Crisp et al., 2004) was launched in 2014 aiming at continuing and improving XCO$_2$ observations from space. OCO-2 is part of the A-train satellite constellation and flies in a sun-synchronous orbit whose

ascending node crosses the equator on 13:36 local time. It measures the solar backscattered radiance in three independent wavelength bands in the spectral regions of the near infrared (NIR) and short wave infrared (SWIR): the O$_2$-A band at around 760 nm, the weak CO$_2$ band at around 1610 nm, and the strong CO$_2$ band at around 2060 nm. OCO-2 is operated in a near-push-broom fashion and has eight footprints across track with a spatial resolution at ground of 1.29 km×2.25 km.

We use NASA's operational bias corrected OCO-2 L2 Lite XCO$_2$ product v9 (Kiel et al., 2018, see Fig. 1a for an example)

which we obtained from *https://daac.gsfc.nasa.gov*. The data set is rigorously pre- and post-filtered for potentially unreliable soundings including, e.g., cloud and aerosol contaminated scenes. Additionally, the OCO-2 retrieval algorithm accounts for light scattering at optically thin aerosol layers by fitting the optical depth and height of two lower-atmosphere aerosol layers and the optical depth of a stratospheric aerosol layer (O'Dell et al., 2018).

The OCO-2 XCO$_2$ product includes an uncertainty estimate which we use for our study. For the selected scenarios, the

15 reported single sounding uncertainty lies typically in the range of 0.4 ppm to 0.7 ppm which is similar to estimates based on the standard deviation of the difference of succeeding soundings. The validation study of Reuter et al. (2017) estimated that the single sounding precision relative to ground based Total Carbon Column Observing Network (TCCON) data is about 1.3 ppm. However, this includes, e.g., the noise of the validation data set and a larger pseudo-noise component due to spatial and temporal representation errors when co-locating OCO-2 with the validation data and it shall be noted that the study of

20 Reuter et al. (2017) analyzed a predecessor NASA OCO-2 XCO$_2$ data set (v7 instead of v9).

### 2.2 NO$_2$

The TROPOspheric Monitoring Instrument (TROPOMI) on Sentinel-5 Precursor was launched in October 2017 into a sun-synchronous orbit with an ascending node local equator crossing time of 13:30 (Veefkind et al., 2012). TROPOMI is a nadir viewing imaging grating spectrometer for the UV/visible spectral region with additional channels in the NIR and SWIR, extend-

25 ing the existing data records of the GOME (Global Ozone Monitoring Experiment), SCIAMACHY, OMI (Ozone Monitoring Instrument), and the GOME-2 missions. It has a swath width of about 2600 km and in comparison to previous instruments a much better spatial resolution of 3.5 km×7 km at nadir at similar signal to noise ratio per measurement. Here we use radiances in the spectral region 425 nm–465 nm to retrieve NO$_2$ slant columns with a standard Differential Optical Absorption Spectroscopy (DOAS) retrieval developed for previous satellite instruments (Richter et al., 2011), followed by a de-striping step as

described by Boersma et al. (2007). Slant columns are defined as the absorber concentration integrated along the light path, and thus depend on both, the atmospheric NO$_2$ profile, and the light path of the individual measurement.





Evaluation of the scatter of viewing angle corrected $NO_2$ slant columns over a clean Pacific region (10°S–10°N, 160°E–230°E) indicates that the random noise ($1\sigma$) of our S5P slant columns is $5 \cdot 10^{14}$ molec./cm$^2$. For individual soundings, the uncertainty can differ depending on viewing geometry and surface reflectance.

In order to extract the tropospheric vertical columns, first the stratospheric contribution to the retrieved slant columns needs to be removed and then the light path dependency of the remaining tropospheric slant columns is corrected for by dividing through a scene dependent air mass factor. In this study, another approach is taken as only localized enhancements are evaluated. By subtracting the surrounding background values (section 2.5), both the stratospheric contribution and any tropospheric background are removed from the signal as they are both smooth on scales of a few tens of kilometers discussed here. What remains is the slant column plume signal of the lower troposphere from which we derive information on the $CO_2$ plume.

## 2.3 Co-location of OCO-2 and S5P data

OCO-2 and S5P fly both in sun-synchronous orbits with similar equator crossing times of their ascending nodes and orbit times of about 100 minutes. S5P has a swath width of about 2600 km which provides nearly global coverage each day. For these reasons, basically each scene observed by OCO-2 is also observed by S5P within a maximum time difference of about 50 minutes. We project the S5P and OCO-2 data of the same day in a surrounding of a potential target on a high resolution (0.001° × 0.001°) grid to compute $NO_2$ averages representative for the footprints of the $CO_2$ soundings (see Fig. 1c for an example).

## 2.4 Geophysical data bases

As input for the computation of the cross-sectional fluxes (section 2.5), we compute the number of dry air particles in the atmospheric column from meteorological profiles which we read at the same time with the wind information from the ECMWF (European Centre for Medium range Weather Forecast) ERA5 (fifth generation of ECMWF atmospheric reanalyzes) data archive at 0.25°×0.25°×hourly resolution. This data archive provides also an uncertainty estimate of the wind information (at three times lower resolution) from an ensemble statistic.

We compare the inferred cross-sectional $CO_2$ fluxes with the following emission data bases. The Emissions Database for Global Atmospheric Research (EDGAR v4.3.2, *https://edgar.jrc.ec.europa.eu*) provides information on anthropogenic $CO_2$ emissions at 0.1°×0.1°×annually resolution. EDGAR v4.3.2 ends in 2012 and we use the data of that year for our comparisons. The Open-Data Inventory for Anthropogenic Carbon dioxide (ODIAC v2017, *http://db.cger.nies.go.jp/dataset/ODIAC*, Oda et al., 2018) provides also information on annual anthropogenic $CO_2$ emissions but at a higher resolution (1 km×1 km×monthly) and the data base ends in 2016. For the reason of comparability, we re-gridded the ODIAC emissions to the EDGAR resolution (0.1°×0.1°×annual) and use 2012 data as baseline. Additionally, we use ODIAC v2017 data re-gridded to 0.1°×0.1°×monthly resolution. The Global Fire Emissions Database (GFED v4.1s, *https://www.globalfiredata.org*) provides information on $CO_2$ emissions from wildfires at a resolution of 0.25°×0.25°×3 hours which we re-gridded to 0.1°×0.1° resolution for a six hours average ending approximately at the time of the overpass.





## 2.5 Flux estimation

S5P's spatial resolution is considerably coarser than that of OCO-2. Consequently for our case studies, we concentrate on plumes which are significantly larger than the swath width of OCO-2. This means that for the selected scenes, OCO-2 sees actually only a cross-section of a plume (see Fig. 1c for an example).

We model the cross-sectional $NO_2$ columns along the OCO-2 orbit by a first degree polynomial (i.e., a linear polynomial) accounting for large scale variations of the background values overlayed by a Gaussian function describing the enhancement within the plume. Simultaneously, the cross-sectional $CO_2$ concentrations are modeled in a similar manner. However, the width of the $CO_2$ Gaussian function is constrained to equal the width of the $NO_2$ Gaussian function. This means, the plume shape is determined from the $NO_2$ measurements, but we allow for a shifted position of the maximum in order to account for potential

plume displacements resulting from different overpass times. Additionally, it shall be noted that the $CO_2$ and $NO_2$ plumes may have small differences, e.g., due to different decay rates of $NO_2$ in different altitudes. These differences, however, are considered minor compared with the precision of the $XCO_2$ soundings. Specifically, the $NO_2$ and $XCO_2$ values along the distance in OCO-2's flight direction $x$ are fitted with the maximum likelihood method by the following vector function:

$$\begin{pmatrix} NO_2 \\ XCO_2 \end{pmatrix} = \begin{pmatrix} a_0 + a_1\,x + a_2\,e^{-4\,ln(2)\,(x-a_3)^2\,a_4^{-2}} \\ a_5 + a_6\,x + a_7\,e^{-4\,ln(2)\,(x-a_8)^2\,a_4^{-2}} \end{pmatrix} \tag{1}$$

    The free fit parameters $a_{0-8}$ correspond to the polynomial coefficients of the background values ($a_{0,1,5,6}$), the amplitudes ($a_{2,7}$), shifts ($a_{3,8}$), and the full width at half maximum (FWHM, $a_4$) of the Gaussian functions. Integration over the Gaussian enhancement results in the cross-sectional $CO_2$ flux $F_{CO_2}$ (in units of $MtCO_2/a$) of the plume depending on the FWHM $a_4$

(in km), the amplitude $a_7$ (in ppm), the effective wind speed $v_e$ (in m/s) within the plume normal to the OCO-2 orbit, and the number of dry air particles in the atmospheric column $n_e$ (in $cm^{-2}$):

$$F_{CO_2} = \frac{1}{2}\sqrt{\frac{\pi}{ln(2)}} \cdot 3600 \cdot 24 \cdot 365\,\frac{s}{a}\,\frac{M_{CO_2} \cdot \frac{10^{-12}Mt}{g}}{N_A} \cdot n_e\,\frac{10^4 cm^2}{m^2} \cdot a_4\,\frac{10^3 m}{km} \cdot a_7\,\frac{10^{-6}}{ppm} \cdot v_e \tag{2}$$

Here, $M_{CO_2}$ is the molar mass of $CO_2$ (in g/mol) and $N_A$ the Avogadro constant. We approximate the number of dry air particles $n_e$ and the effective wind speed's normal $v_e$ from ECMWF ERA5 meteorological profiles at the position of the maximum of the fitted Gaussian $XCO_2$ function. In regions with large variations of the surface elevation or wind conditions within the plume's cross-section, it might be appropriate to account for variations in the number of dry air particles and/or the wind conditions when integrating over the Gaussian enhancement. We manually adjust the ECMWF wind direction (not the



wind speed) to subjectively fit the plume direction observed in the $NO_2$ fields (e.g., Fig. 1a). For a hydrostatic atmosphere with a standard surface pressure of 1013hPa, Eq. 2 becomes approximately:

$$F_{CO_2} \approx 0.53 \, \frac{MtCO_2}{a} \, a_4 \, a_7 \, v_e \tag{3}$$

Varon et al. (2018) proposed to approximate the effective wind speed within the plume from the 10 m wind by applying a multiplier in the range of 1.3–1.5. Therefore, we decided to use a multiplier of 1.4 for convenience. This empirical relationship accounts, e.g., for plume rise and mixing into altitudes with larger wind speeds. For the present, we consider this approximation adequate for this first study, but we recognize that uncertainties resulting from our estimate of the effective wind speed's normal may be reduced in the future by improved wind knowledge. However, in this study the focus is on demonstrating the benefits
of simultaneous $NO_2$ and $XCO_2$ measurements rather than on most accurate flux estimates.

Additionally, it shall be noted that the plume cross-sectional flux (Eq. 2) is only a good approximation for the actual source emission under steady state conditions for wind speeds greater than about 2 m/s (Varon et al., 2018) when advection dominates over diffusion (Sharan et al., 1996). Changes in wind direction, wind speed, or atmospheric stability in the time span between emission and observation may result in differences between the plume cross-sectional flux and the source flux. Temporal
variations in the source emissions of course also result in (temporally delayed) variations of the plume cross-sectional flux, which is always only a snap shot and must not be confused with, e.g., the annual average, even though given in the same units. In case of chemically active species (such as $NO_2$), also chemical processes along the plume path would have to be considered in order to compute source emissions from plume cross-sectional fluxes.

## 2.6  Uncertainty propagation

In order to estimate the uncertainty of the $CO_2$ plume cross-sectional flux ($F_{CO_2}$, Eq. 2), we propagate the uncertainties of the FWHM ($a_4$), the amplitude ($a_7$), and the wind speed normal ($v_e$) by assuming uncorrelated errors. The uncertainties of the FWHM and the amplitude result from the maximum likelihood fitting method propagating the uncertainties of the individual $XCO_2$ and $NO_2$ soundings as reported in the data products. The uncertainties of the wind components are read from the ECMWF ERA5 data archive. Additionally, we assume that the manual adjustment of the wind direction is accurate by $\pm 10°$.
These uncertainties propagate into the uncertainty of the wind speed normal. Varon et al. (2018) estimated that computing the effective wind speed from the 10 m wind introduces an additional uncertainty of 8-12%. However, we analyze scenes with larger plume structures and probably also larger variations of the injection heights which we consider by enhancing this error component to 20% for convenience. Uncertainties in the number of dry air particles are neglected as they are much smaller compared to, e.g., the wind speed uncertainty. As mentioned earlier, the assumption of constant meteorological conditions
might not be valid in regions with large variations of the surface elevation or wind conditions within the plume's cross-section, which may result in an underestimation of the total cross-sectional flux uncertainty in such cases.





## 2.7 Plume detection and scenario selection

We use a semi-automatic method to select potentially interesting targets. In a first step, all co-locations of OCO-2 and S5P are computed similarly as described in section 2.3 but based on a coarser high resolution grid ($0.01° \times 0.01°$) to improve the computational efficiency. We shift a $30\,s$ ($\sim 200\,km$) search window in time steps of $0.25\,s$ ($\sim 2\,km$) over the time series of

co-locations. Only those time steps are further considered which have at least 100 co-locations without data gaps exceeding $3\,s$ ($\sim 20\,km$) within the search window. In the next step, we perform a least squares fit of the co-located $XCO_2$ and $NO_2$ data with a Gaussian vector function. This fitting function corresponds to Eq. 1 but with independent FWHM for $XCO_2$ and $NO_2$ and centered within the search window ($a_3$ and $a_8$ set to zero), which improves the convergence rate. Only those time steps are further considered fulfilling the following criteria: the fit converged, the $NO_2$ amplitude exceeds $10^{15}$ molec./cm$^2$, the $XCO_2$

and $NO_2$ FWHM ($a_c$ and $a_n$, respectively) do not exceed the half width of the search window ($a_c, a_n \leq 15\,s$) and do not differ by more than their average ($|a_c - a_n| \leq (a_c + a_n)/2$), the $XCO_2$ and $NO_2$ amplitudes are at least two times larger than their uncertainties and larger than the maximum variations of the backgrounds. In the last step, we decided by manual inspection of the $XCO_2$ and $NO_2$ co-locations plus the surrounding $NO_2$ fields and ECMWF wind information if the scenario is a promising candidate for further flux analyses. Potential reasons to reject an automatically pre-selected scene are, e.g., too low wind speed,

wind direction nearly parallel to OCO-2 orbit, unclear source attribution, or poor fit quality. In total we manually selected about 20 promising scenes in the time period 03/2018 to 08/2018, of which we show 6 examples here.

## 3   Results

From the time period of 03/2018 to 08/2018, we selected the following scenarios as examples for flux analyses based on co-located $XCO_2$ and $NO_2$ observations.

### 3.1   Moscow

Fig. 1a shows the $NO_2$ enhancement in Moscow's city plume as retrieved from S5P overlayed by OCO-2's $XCO_2$ measurements. The $NO_2$ enhancement is clearly visible also in the plume's cross-section along OCO-2's ground track (Fig. 1c). Due to the larger relative noise of the $XCO_2$ retrievals, the $XCO_2$ enhancement is less obvious but still visible (Fig. 1c). The Gaussian fit of the enhancements is excellent for $NO_2$ and reasonable ($\chi^2 = 2.2$) for $XCO_2$. There is only a small adjustment needed

to bring the ECMWF 10 m wind in good agreement with the $NO_2$ plume (Fig. 1a). The effective wind speed normal to the OCO-2 orbit amounts to $1.6 \pm 0.6$ m/s which is a bit lower than optimal for reasonable flux estimates (Varon et al., 2018). The cross-sectional $CO_2$ flux amounts to $76 \pm 33$ MtCO$_2$/a. This compares to 2012 average upwind emissions (white marked boxes in Fig. 1a) of 195 MtCO$_2$/a (EDGAR) and 102 MtCO$_2$/a (ODIAC). ODIAC's emission estimate for 08/2016 amounts to 88 MtCO$_2$/a. The $NO_2$ far field shows no indications for overlayed $CO_2$ plumes from other sources (Fig. 1b). The total flux

uncertainty is dominated by the uncertainty of the wind direction followed by the uncertainty of the effective wind speed.



**Table 1.** Summary of cross-sectional flux results including uncertainty contributions ($1\sigma$) and comparison with emission data bases EDGAR and ODIAC or GFED in the case of the Australian wildfires. The ODIAC values in brackets represent ODIAC emissions of 2016 and the month of the overpass in the same grid boxes as summed up for 2012. The uncertainty estimate comprises the total uncertainty and the uncertainties introduced by the ECMWF wind uncertainty, the uncertainty of the wind direction ($10°$), use of the $10\,m$ wind ($20\%$), the $XCO_2$ precision, and the $NO_2$ precision. All values are in units of $MtCO_2/a$.

| Emission source | Cross-sect. flux | Cross-sectional flux uncertainty | | | | | | EDGAR | ODIAC/ GFED* |
| --- | --- | --- | --- | --- | --- | --- | --- | --- | --- |
| | | Total | ECMWF | Angle | 10 m | $XCO_2$ | $NO_2$ | | |
| Moscow | 76 | 33 | 4 | 29 | 15 | 5 | 1 | 195 | 102 (88) |
| Lipetsk | 69 | 50 | 5 | 48 | 14 | 1 | 0 | 23 | 4 (4) |
| Baghdad | 95 | 36 | 3 | 30 | 19 | 6 | 1 | 22 | 13 (12) |
| Medupi and Matimba | 31 | 7 | 3 | 2 | 6 | 2 | 0 | 0 | 24 (26) |
| Australian wildfires* | 153 | 40 | 5 | 24 | 31 | 8 | 5 | 0 | 52 |
| Nanjing | 120 | 27 | 10 | 5 | 24 | 6 | 1 | 164 | 89 (96) |

## 3.2 Lipetsk

Fig. 2a shows the surrounding of Lipetsk with the Novolipetsk steel plant and the Lipetskaya TEC-2 gas-fired power plant (515MW) only one minute ( 400 km) apart from Moscow along OCO-2's flight track (see also Fig. 1b). The cross-sectional $NO_2$ and $XCO_2$ enhancements clearly stand out of the noise of the data (Fig. 2c) and the Gaussian function fits the $XCO_2$ data

reasonably well ($\chi^2 = 2.4$). We applied only a small correction to the ECMWF wind direction. However, as the wind direction is similar to OCO-2's flight direction, the normal effective wind speed is unfavorably low ($0.9\pm0.7\,m/s$) which makes the cross-sectional flux estimates ($69\pm50\,MtCO_2/a$) less reliable and highly uncertain. The by far larges uncertainty contribution comes from the uncertainty of the wind direction. The 2012 average EDGAR and ODIAC upwind emissions (white marked boxes in Fig. 2a) are $23\,MtCO_2/a$ and $4\,MtCO_2/a$ (same for 08/2016), respectively, but the $NO_2$ far field shows no indications

for overlayed $CO_2$ plumes from other sources (Fig. 2b).

## 3.3 Baghdad

Fig. 3a shows the S5P $NO_2$ slant columns overlayed by OCO-2 $XCO_2$ data in a surrounding of Baghdad. Enhanced values are clearly visible in the cross-section of the $NO_2$ plume and less obviously visible also in the $XCO_2$ data (Fig. 3c). The $XCO_2$ enhancement is well fitted ($\chi^2 = 1.0$) by the Gaussian fitting function. The manually adjusted wind direction is similar

to the ECMWF wind direction and the normal wind speed amounts to $4.4\pm1.7\,m/s$. From the $XCO_2$ enhancement and the normal wind speed, we compute the cross-sectional $CO_2$ flux to be $95\pm36\,MtCO_2/a$. This compares to an upwind source of $22\,MtCO_2/a$ or $13\,MtCO_2/a$ ($12\,MtCO_2/a$ for 07/2016) of EDGAR or ODIAC, respectively. The flux uncertainty is dominated by the uncertainty of the wind direction and the uncertainty of the effective wind speed. The $NO_2$ far field shows no indications for overlayed $CO_2$ plumes from other sources (Fig. 3b).




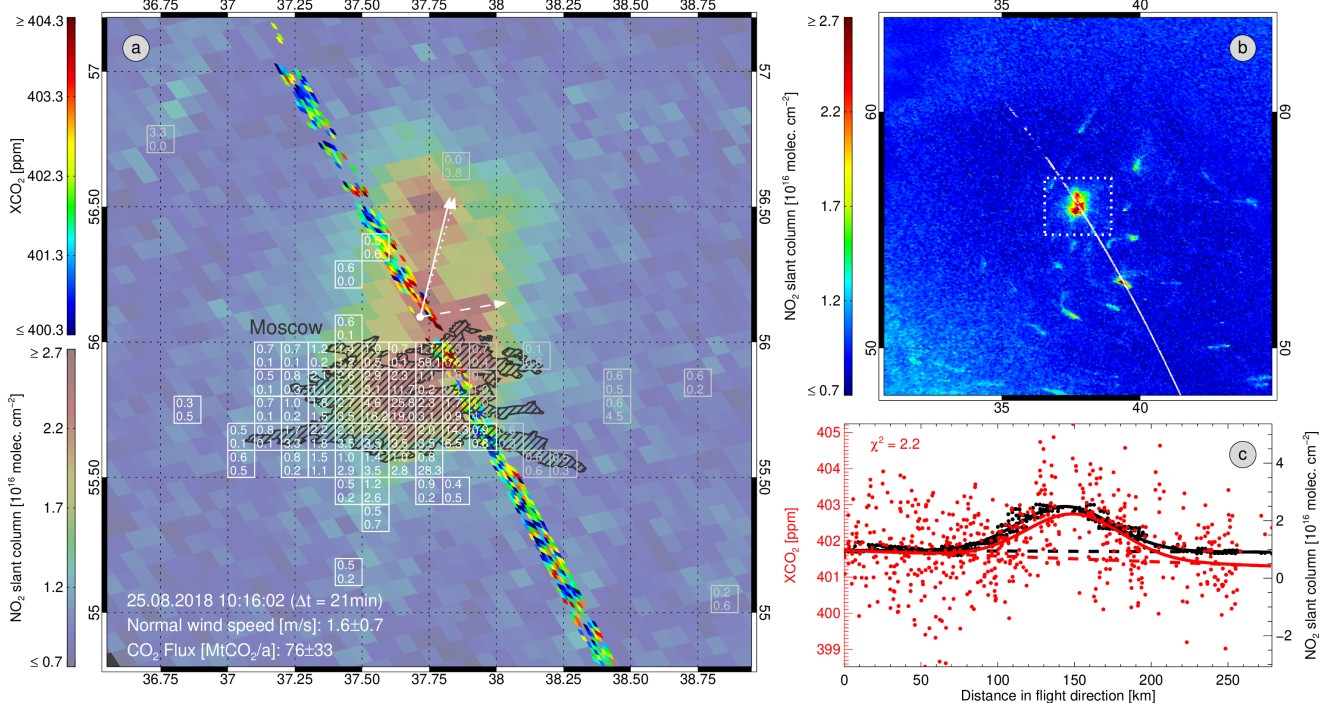

**Figure 1.** Moscow on August 25, 2018. **a)** S5P $NO_2$ slant column (background) overlayed by OCO-2 $XCO_2$ (foreground). Gray and white $0.1°$ boxes show EDGAR (bottom) and ODIAC (top) 2012 annual emissions with either EDGAR or ODIAC being larger than $0.5\,MtCO_2/a$. The white arrows show the direction of the 10 m wind as read from ECMWF (dotted), manually corrected to (subjectively) best match the $NO_2$ plume (solid), and normal to the OCO-2 orbit (dashed). Effective wind speed normal to the OCO-2 orbit, estimated cross-sectional $CO_2$ flux, time of OCO-2 overpass, and time difference between OCO-2 and S5P overpass are also listed. **b)** Larger section of the S5P $NO_2$ slant columns including the OCO-2 orbit and the bounding box of sub-figure a). **c)** OCO-2 $XCO_2$ values (red) and co-located S5P $NO_2$ slant columns (black) within the plume's cross-section in OCO-2 flight direction.

## 3.4 Medupi and Matimba power plants

The Medupi (4764MW) and Matimba (3990MW) coal-fired power plants lie close to each other in South Africa about 300 km north of Johannesburg. Their $NO_2$ plume is shown in Fig. 4a overlayed by $OCO_2$ $XCO_2$ measurements. $NO_2$ measurements in the larger surrounding do not suggest any additional nearby upwind sources (Fig. 4b). The cross-sectional $NO_2$ values

5    show a clear elevation within the plume which is less obvious for $XCO_2$ having larger relative scatter especially south of the plume. Nevertheless, the Gaussian function fits the $XCO_2$ values reasonably well ($\chi^2 = 1.4$). The wind direction is nearly perpendicular to the OCO-2 orbit and the effective normal wind speed is 2.6±0.6 m/s. The cross-sectional $CO_2$ flux amounts to $31\pm7\,MtCO_2/a$ which is consistent with ODIAC 2012 emissions of $24\,MtCO_2/a$ and ODIAC 07/2016 emissions of $26\,MtCO_2/a$ but EDGAR does not have significant emissions in this area. It shall be noted that the Medupi power plant started operation

10   in 2015 with limited capacity and that it still has not reached its nominal capacity. Therefore, it is no surprise that the Medupi





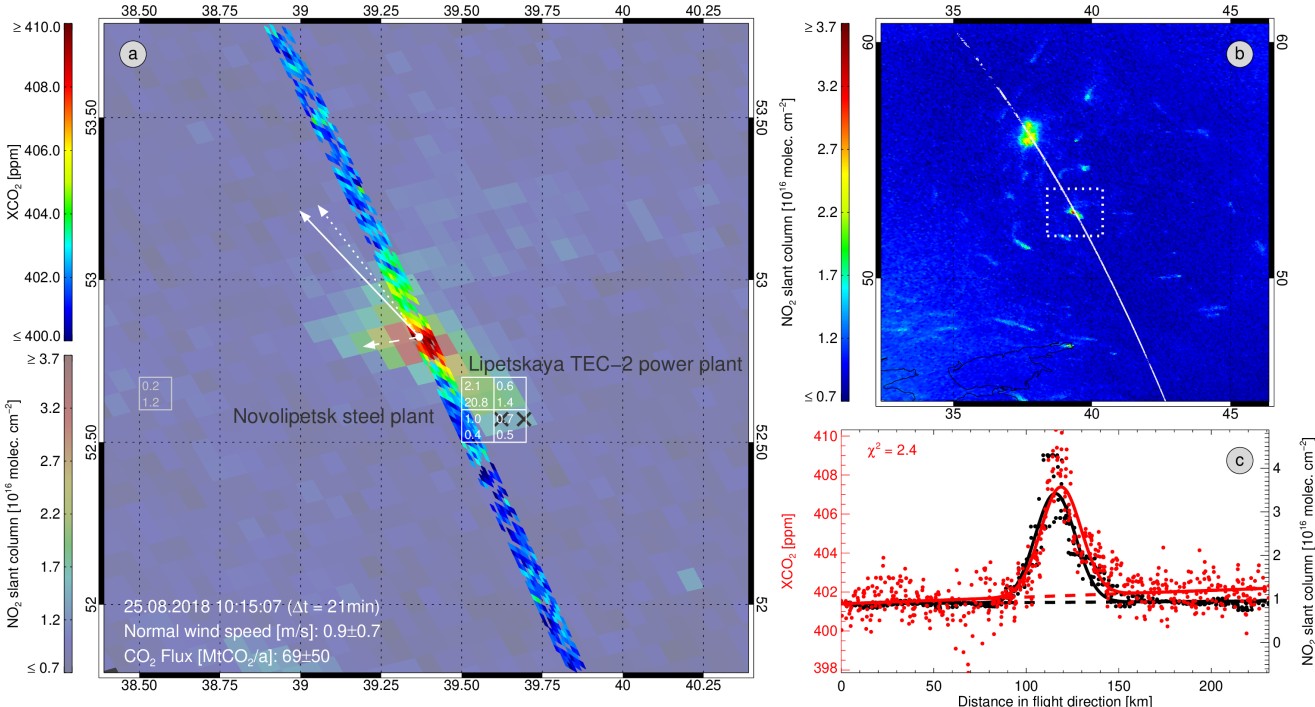

**Figure 2.** As Fig. 1 but for Lipetsk on August 25, 2018.

power station is not included in either EDGAR or ODIAC 2012 data. The flux uncertainty is dominated by the uncertainty of the effective wind speed.

### 3.5 Australian wildfires

Fig. 5a shows the $NO_2$ plumes of two Australian wildfires on 05.05.2018 overlayed by an OCO-2 orbit of $XCO_2$ measurements.

5   Enhanced $NO_2$ and $XCO_2$ values are clearly visible within the plume's cross-section (Fig. 5b). The $NO_2$ (and less obvious also the $XCO_2$) cross-section has two maxima. The Gaussian fitting function cannot account for this, which is, however, not reflected in the overall good fit quality ($\chi^2 = 0.6$). We applied only a small manual correction to the wind direction and the effective wind speed normal to the OCO-2 orbit is 6.7±1.7 m/s. For the snapshot of the overflight, we computed a cross-sectional $CO_2$ flux of 153±40 $MtCO_2$/a. Its uncertainty is driven by the uncertainty of the effective wind speed and wind direction. As the shown

10   plumes originate from wildfires, EDGAR and ODIAC do not include their emissions. However, GFED has average emissions of 52 $MtCO_2$/a within the six hours period 0h–6h UTC including the time of the overpass (5h UTC). The maximum GFED emissions are approximately at the position of the largest $NO_2$ concentrations. Fig. 5c shows no indications, that additional upwind sources may explain the discrepancy between our cross-sectional flux estimate and GFED.





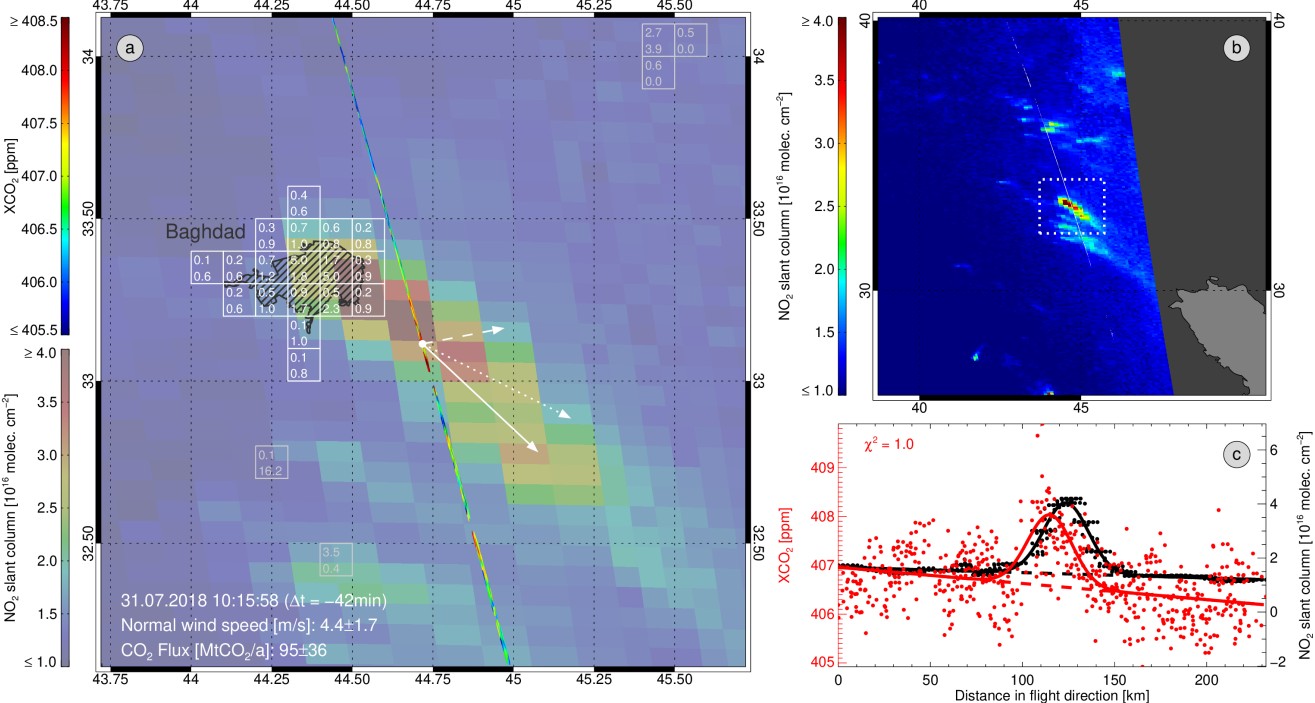

**Figure 3.** As Fig. 1 but for Baghdad on July 31, 2018.

### 3.6 Nanjing

Fig. 6a shows the $NO_2$ slant columns in the surrounding of Nanjing overlayed by OCO-2 $XCO_2$ measurements. The cross-section along the OCO-2 orbit shows strong $XCO_2$ and and $NO_2$ plume signals distinctively above the noise level which are well fitted with the Gaussian fitting function ($\chi^2 = 0.6$). The ECMWF wind direction is not far from being rectangular to the OCO-2 orbit and we applied only a moderate manual correction. The effective normal wind speed is $2.2\pm0.5$ m/s. This results in a cross-sectional flux estimate of $120\pm27$ $MtCO_2$/a which lies in between the upwind emissions of EDGAR ($163$ $MtCO_2$/a) and ODIAC ($89$ $MtCO_2$/a for 2012, $96$ $MtCO_2$/a for 03/2016). Fig. 6b does not indicate additional major remote upwind sources. The uncertainty of the cross-sectional flux estimate is dominated by the uncertainty of the effective wind speed.

### 4 Summary and conclusions

Based on six case studies (Moscow, Russia; Lipetsk, Russia; Baghdad, Iraq; Medupi and Matimba power plants, South Africa; Australian wildfires; and Nanjing, China), we demonstrated the usefulness of simultaneous satellite observations of $NO_2$ and the column-average dry-air mole fraction of $CO_2$ ($XCO_2$). For this purpose, we analyzed co-located regional enhancements of $XCO_2$ observed by OCO-2 and $NO_2$ observed by S5P and estimated the $CO_2$ plume's cross-sectional fluxes. For atmospheric





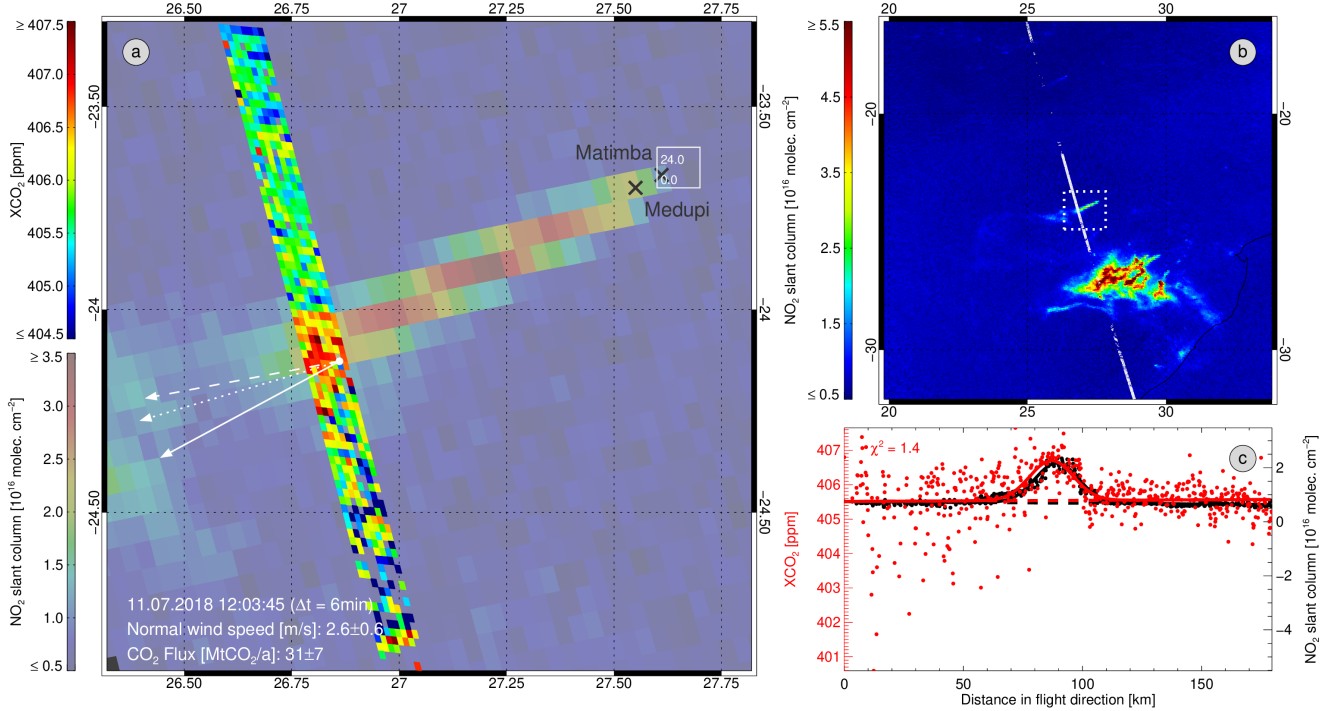

**Figure 4.** As Fig. 1 but for the Medupi and Matimba power plants in South Africa on July 11, 2018.

standard conditions, we approximated as a rule of thumb, that a Gaussian enhancement of 1 ppm with a width of 1 km at a wind speed (normal to the cross-section) of 1 m/s corresponds to a plume cross-sectional flux of roughly 0.53 MtCO$_2$/a.

For Moscow, we derived a cross-sectional flux of $76\pm33$ MtCO$_2$/a which agrees (within its uncertainty) with ODIAC 2012 emissions of 102 MtCO$_2$/a (88 MtCO$_2$/a for 08/2016) but not with EDGAR emissions of 195 MtCO$_2$/a. The cross-sectional

flux estimate of Lipetsk with the Novolipetsk steel plant and the Lipetskaya TEC-2 power plant is $69\pm50$ MtCO$_2$/a. Within its uncertainty, this estimate agrees with EDGAR emissions of 23 MtCO$_2$/a but not with ODIAC emissions of 4 MtCO$_2$/a. However, the uncertainty of the estimate is large due to a wind direction with an acute angle relative to the OCO-2 orbit which also results in a low effective normal wind speed. This can serve as an example for low wind speeds being favorable for plume detection but not necessarily for flux quantification. In the case of Baghdad, we derived a cross-sectional flux of $95\pm36$ MtCO$_2$/a for

the time of the overpass which is considerably larger than the annual average EDAGR (22 MtCO$_2$/a) and ODIAC (13 MtCO$_2$/a for 2012, 12 MtCO$_2$/a for 07/2016) emissions of 2012. The wind conditions were relatively good and S5P NO$_2$ measurements do not suggest an overlaying significant upwind source. In this context, it is interesting to note that Georgoulias et al. (2018) found a strongly increasing trend ($17.0\pm0.8\%$/a in the period 04/1996–09/2017) for the tropospheric NO$_2$ concentrations in Baghdad (and a decreasing trend of -2.2$\pm$0.7%/a for Iraq) hinting at strongly increasing CO$_2$ emissions in Baghdad since

2012. The cross-sectional flux of the plume of the Medupi and Matimba power plants have been estimated to $31\pm7$ MtCO$_2$/a which agrees (within its uncertainty) with ODIAC (24 MtCO$_2$/a for 2012, 26 MtCO$_2$/a for 07/2016) but not with EDGAR (no



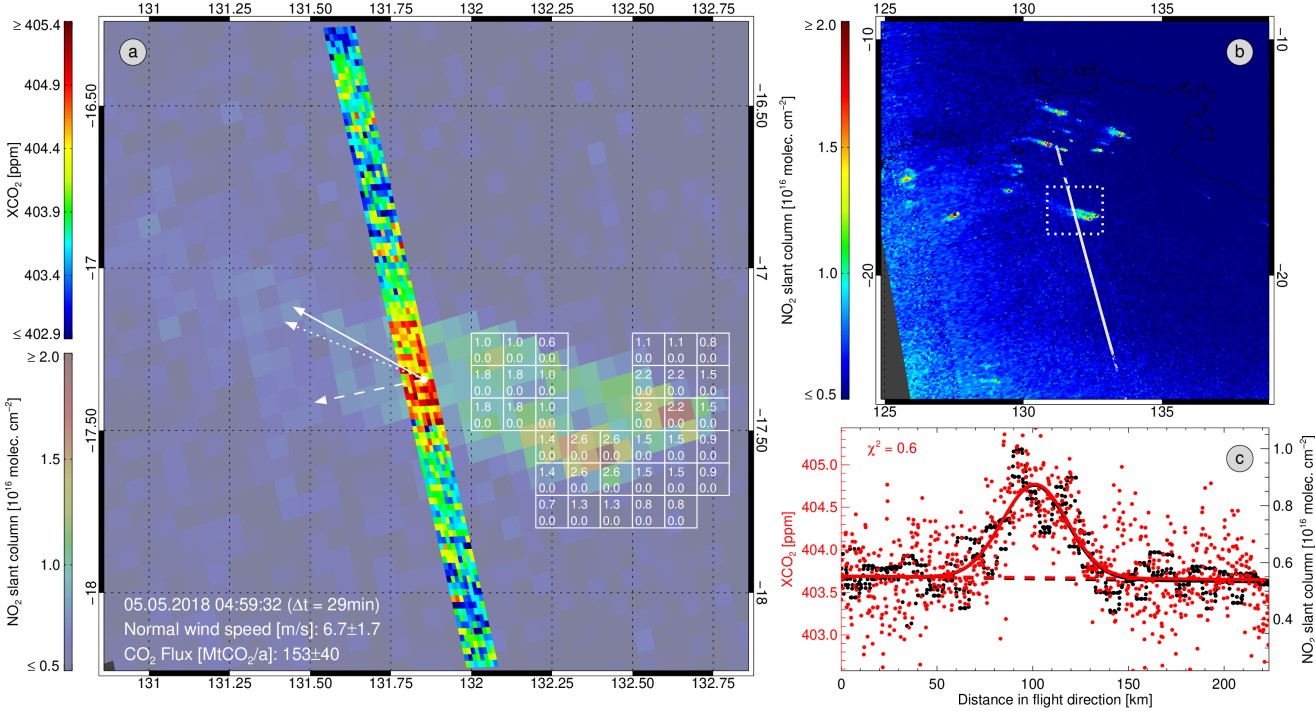

**Figure 5.** As Fig. 1 but for Australian wildfires on May 5, 2018. The ODIAC emission data (top number) have been replaced by GFED emissions for the time of the OCO-2 overpass.

significant emission). For the Australian wildfires, we estimated a plume cross-sectional flux of $153\pm40\,\text{MtCO}_2/\text{a}$ which is about three times larger than the GFED estimate ($52\,\text{MtCO}_2/\text{a}$) for a six hours average ending approximately at the time of the OCO-2 overflight. Unfavorable wind conditions or a strong overlaying upwind source can be excluded as reason for the discrepancy. The same is true for the fact that a double-plume structure has been fitted with a Gaussian function. However,

it shall be noted that GFED's emission estimate for the same time interval but one day before the OCO-2 overpass amounts to $252\,\text{MtCO}_2/\text{a}$. For the Nanjing scene, we derived a cross-sectional flux of $120\pm27\,\text{MtCO}_2/\text{a}$ which lies in between ODIAC ($89\,\text{MtCO}_2/\text{a}$ for 2012, $96\,\text{MtCO}_2/\text{a}$ for 03/2016) and EDGAR ($164\,\text{MtCO}_2/\text{a}$). However, the scene includes a larger area of overlayed sources, making source attribution difficult.

The total uncertainty of the derived plume cross-sectional fluxes ranges from $7\,\text{MtCO}_2/\text{a}$ to $50\,\text{MtCO}_2/\text{a}$ or in relative mea-

sures from 23% to 72%. The total uncertainty is always dominated by an uncertainty contribution related to meteorology. Specifically, the (manually adjusted) wind direction or the computation of the effective wind speed from the 10 m wind contribute most to the total uncertainty. The noise of the $XCO_2$ retrievals contributes only with $1\,\text{MtCO}_2/\text{a}$ to $8\,\text{MtCO}_2/\text{a}$ to the total error and the noise of the $NO_2$ retrievals contributes on average even three times less.

It is unlikely, that the observed $XCO_2$ enhancements are dominated by uncorrected enhancements due to co-emitted aerosols

because the OCO-2 retrieval algorithm accounts for light scattering at optically thin aerosol layers and filters scenes with





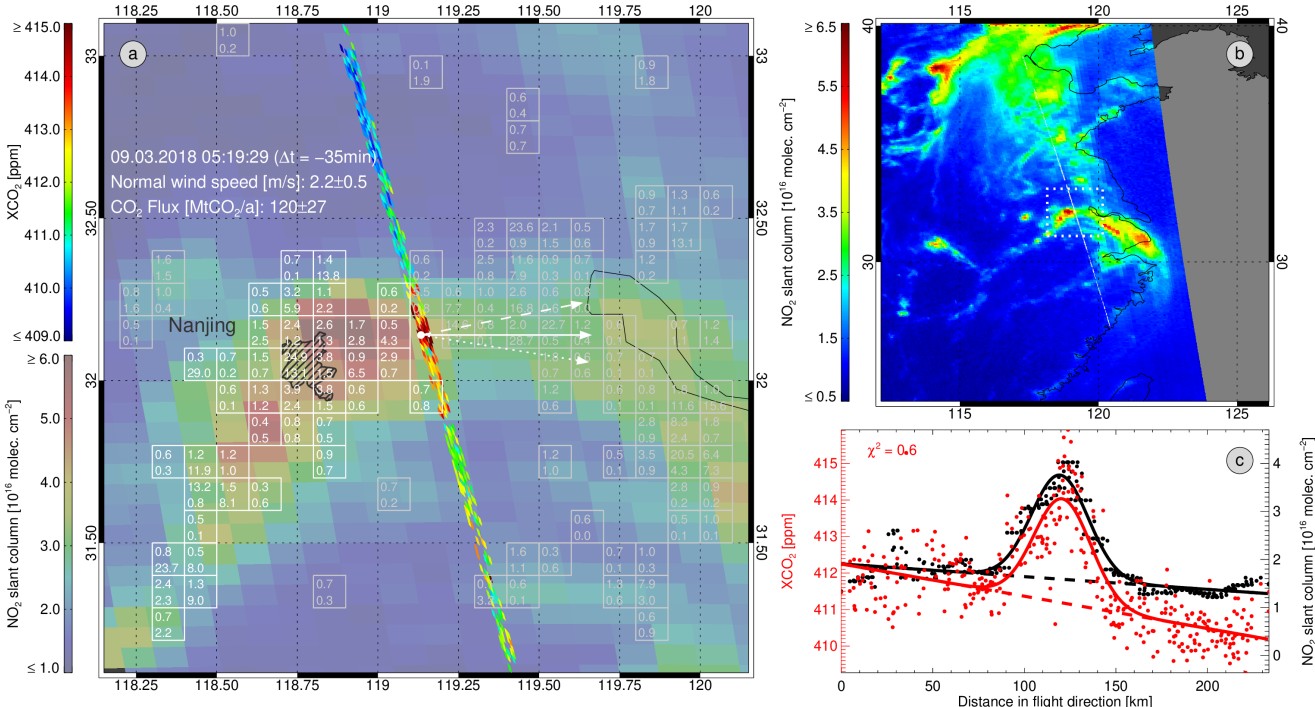

**Figure 6.** As Fig. 1 but for Nanjing on March 9, 2018.

stronger aerosol contamination. Additionally, Bovensmann et al. (2010) estimated for the proposed CarbonSat (Carbon Monitoring Satellite) instrument that neglecting co-emitted aerosols in power plant plumes results in errors between $0.2\,\mathrm{MtCO_2/a}$ and $2.5\,\mathrm{MtCO_2/a}$ which is small compared with the derived cross-sectional fluxes and their total uncertainties (Tab. 1). Aerosols can also effect the S5P $NO_2$ slant columns which is, however, less important for our work because we derive only the plume

width and direction from the $NO_2$ observations.

  It shall be noted that differences of the cross-sectional flux estimates and the emission data bases are not necessarily coming from inaccuracies of the satellite retrievals or the emission data bases. Our estimates are valid only for the time of the overpass while the emission data bases give annual or monthly averages. Velazco et al. (2011) illustrated, that power plants can have substantial annual and day-to-day variations. Additionally, the cross-sectional flux is only a good approximation for the source

emission under meteorological steady state conditions with wind speeds greater than about 2 m/s (Varon et al., 2018).

  For the analyzed scenarios, we observe rather large differences between the EDGAR and ODIAC emission inventories. However, note that only those grid boxes are shown (and summed up) in Fig. 1a–6a for which either EDGAR or ODIAC emissions are larger than $0.5\,\mathrm{/MtCO_2/a}$. This means, a smoother distribution of emissions may be misinterpreted as less emissions, if a significant fraction of the total emission is located in grid boxes not exceeding the $0.5\,\mathrm{/MtCO_2/a}$ threshold.

Additionally, it shall be noted that ODIAC emissions correspond to fossil fuel combustion and cement production only, while EDGAR includes also emissions from other sectors (e.g., agriculture, land use change, and waste).



NO$_2$ is co-emitted with CO$_2$ when fossil fuels are combusted at high temperatures and has a relatively short lifetime of the order of hours which makes it a suitable tracer for recently emitted CO$_2$. We take advantage of this by using NO$_2$ measurements to i) identify the source of the observed XCO$_2$ enhancements, ii) to exclude interference with potential additional remote upwind sources, iii) to manually adjust the wind direction, and iv) to put a constraint on the shape of the observed CO$_2$ plumes.

5     In principle, it is also be possible, to fit only the XCO$_2$ values without constraining the plume shape by NO$_2$. In this case, XCO$_2$ is used to derive the amplitude and FWHM of the enhancement. We repeated the flux estimation of all shown scenarios with such a setup and got consistent flux results but with an uncertainty contribution due to the noise in the XCO$_2$ data increased by about 33%.

    Reuter et al. (2014) discussed that post ENVISAT missions such as OCO-2 would benefit from co-located measurements 10  of co-emitted species from other satellites or ideally multi-species measurements from the same instrument. We demonstrated, that the analysis of small scale emissions in OCO-2 XCO$_2$ data indeed profits from simultaneous NO$_2$ observations of S5P as they allow to set the XCO$_2$ observations into context but also to constrain the plume structure. The uncertainties of the cross-sectional flux estimates due to meteorology and their agreement with the actual emissions might be improved in subsequent studies by making use of dedicated simulations with Lagrangian particle dispersion models with either known source positions 15  (and injection heights) or source positions inferred from the NO$_2$ data.

    However, we expect the largest room for improvements in satellite missions such as the planned European Copernicus anthropogenic CO$_2$ monitoring mission (CO2M) which will provide not only precise measurements with high spatial resolution but also imaging capabilities with a wider swath of simultaneous XCO$_2$ and NO$_2$ observations. Its imaging capabilities will reduce the uncertainty of the inferred emissions due to measurement noise simply because of the increased number of sound-20  ings. Additionally, simultaneous XCO$_2$ and NO$_2$ observations from the same platform will allow stricter constraints on the plume shape. More importantly, the meteorology related uncertainties will reduce (Varon et al., 2018) because deviations from steady state conditions can average out and are, therefore, less critical if the entire plume structure is sampled rather than only a cross-section.

*Author contributions.* M.R.: experimental set-up, data analysis, interpretation, writing the paper. M.B., O.S., S.K., H.B., J.P.B.: experimental 25  set-up, interpretation, improving the paper. A.R.: interpretation, design and operation of the NO$_2$ satellite retrieval, improving the paper. C.W.O'D.: interpretation, design and operation of the XCO$_2$ satellite retrieval, improving the paper.

*Competing interests.* The authors declare that they have no conflict of interest.

*Acknowledgements.* This work was funded by the State and the University of Bremen. The OCO-2 XCO$_2$ data were produced by the OCO-2 project at the Jet Propulsion Laboratory, California Institute of Technology, and obtained from the OCO-2 data archive maintained at





the NASA Goddard Earth Science Data and Information Services Center. S5P is an ESA mission implemented on behalf of the European Commission. The TROPOMI payload is a joint development by ESA and the Netherlands Space Office (NSO). The S5P ground-segment development has been funded by ESA and with national contributions from The Netherlands, Germany, and Belgium. ERA5 meteorological profiles have been obtained from the Copernicus Climate Change Service (C3S) operated by ECMWF. $CO_2$ emission data have been obtained

5    from the EDGAR, ODIAC, and the GFED data bases.





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
