# Peer review of "Towards monitoring localized CO2 emissions from space: co-located regional CO2 and NO2 enhancements observed by the OCO-2 and S5P satellites"

_Atmospheric Chemistry and Physics, 2019_

## Referee Comment (RC1) · Anonymous Referee #2 · 12 Apr 2019

**General comments**

The authors estimate CO2 emissions of cities, power plants and a wild fire from OCO-2 XCO2 observations using a mass-balance approach. They include Tropomi NO2 (slant) columns to improve the emission estimation and present six examples for which emissions were estimated. The paper describes relevant and new ideas and is generally well structured, but more results should be included to support the conclusions better.

[Figure]

Therefore, I recommend a major revision of the manuscript which should address the comments below:

* The number of cases studied is very small. Although 20 promising scenes were identified only six are shown in the manuscript. The authors should include the remaining cases, not as additional examples (except maybe in the supplement), but in order to have more cases to discuss and compare the results.

* The manuscript lacks a detailed comparison of the examples and discussion of the result. A broad summary is given in the conclusions, but this should be moved to a designated section. Furthermore, the results of the flux estimates without including NO2 fields in the fit should be shown in the results and not only briefly mentioned in the conclusions (P15, L6ff).

* The connection between this study and the proposed CO2M mission, which is emphasized in the abstract and the conclusions, is not well presented. The authors used the NO2 fields to identify the location of the source outside the OCO-2 swath and to screen for potential sources upstream. Both will not be possible with the CO2M mission, if CO2 and NO2 instrument would have the same swath as currently proposed. In addition, it might also not be necessary for CO2M to use NO2 to identify the source outside the swath, because CO2M's swath will be significantly wider than OCO-2's swath. A major advantage of the NO2 observations is likely the potential for improving the fit of the Gaussian (see previous comment), which should be presented more prominently.

**Specific comments**

P2, L5ff: Consider re-formulating, e.g.: "... to halve [...] emissions each decade after reaching peak emissions in 2020"

P2, L21-23: Consider to remove, because the detailed chemistry seems not relevant for the study.

P2, L27: Add the resolution of the NO2 instrument.

P2, L29: Clarify the term "localized small scale".

P3, L10f: Consider changing "The data set..." to "The product..." to make it clear that the filtering is part of the OCO-2 L2 Lite product.

P4, L1: Please explain the term "viewing angle correct". Are these geometric air mass factors?

P4, L4-9: The paragraph might be easier readable, if it first explains the approach used in this study and briefly contrasts it the "normal" approach.

P4, L13: It would be useful to have the time difference between OCO-2 and Tropomi for the six examples presented in the manuscript.

P4, L22: Please specify if three times lower resolution is temporal, spatial or both and, if spatial resolution, if it is grid cell area or length. Maybe it's better to just write the resolution of the product.

P5, L7-8: This is not a constraint on the width of the CO2 Gaussian function, if CO2 and NO2 values are fitted simultaneously, but using the same width for fitting both CO2 and NO2 Gaussian function. It would be a constraint if the NO2 plume is fitted first and afterwards the CO2 plume is fitted using the coefficient obtained from the NO2 fit.

P5, L13: Please clarify if a Level-2 or Level-3 product is used for the fit.

P5, L22: The equation would be more readable without the unit conversions. The equation could be split in two.

P5, L27: Does the OCO-2 product have not air columns that could be used instead?

P6, L3: Please add how sensitive the factor (i.e. 0.53) is to surface pressure, to provide a range in which this approximation might be used.

P6, L24: Please state typical values for the uncertainty of the wind speed in the ERA5 data.

P7, L5: Are the 100 co-locations for Level-2 or Level-3?

P15, L1ff: The major advantage of the NO2 instrument here would be the wider swath and thus having CO2 instrument with a wider swath should bring the same advantage without the need for an additional NO2 instrument. The authors should consider discussing if (or why not) having a CO2 instrument with a wide swath is (not) an option.

P15, L22: Please explain the term "steady state conditions" in the paper.

Figure 1-6: The corrected wind arrow looks very subjectively. It might help to draw the arrow centered on the XCO2 swath where the arrow should be a perpendicular to the plume.

Table 1: Mention that the single sounding uncertainty (0.4-0.7 ppm) provided the OCO-2 product was used for computing the flux uncertainty, because evaluation with TCCON found a significant larger value (1.3 ppm).

Table 1: Add to the columns that estimated cross sections are instantaneous; EDGAR and ODIAC are annual or monthly emissions.

**Technical corrections**

P1, L10: XCO2 was already defined in line 3

P5, L5: Consider changing "...first degree polynomial (i.e. a linear polynomial)..." to "...a linear polynomial...".

P5, L6: Consider adding commas before "accounting" and after "values".
* * *

---

## Referee Comment (RC2) · Ray Nassar (Referee) · 19 Apr 2019

Reuter et al. investigate the ability to estimate CO2 emissions from localized sources using satellite observations of CO2 from OCO-2, with the help of NO2 data from Sentinel 5P TROPOMI. The scientific significance of this paper is high since it attempts to address a question of importance to the design of the planned European Copernicus Anthropogenic CO2 Monitoring mission constellation and observations from other existing, planned or proposed satellite missions. Although there have been some past theoretical studies on this subject, this is the first study, to my knowledge, to estimate

local-scale CO2 emissions with real satellite observations of both CO2 and NO2.

The methodology applied wisely uses NO2 (with a wider observational field and shorter atmospheric lifetime than CO2) to effectively identify the plume shape, wind direction and potential interfering sources, and thus the approach is independent of any assumptions about NO2:CO2 emission ratios. The figures and general presentation of the manuscript are of high quality. Overall, this work is sufficient to demonstrate the value of coincident NO2 and CO2 satellite observations for estimating emissions, however, some over-simplifications in the method make the actual emission estimates questionable despite their large uncertainties. These issues include assumptions about the effective altitude of emissions, atmospheric stability, treatment of area sources, and reporting in annual units. All of these issues are elaborated on in the specific points below.

Furthermore, some questions remain unanswered such as an assessment of the impact of a temporal offset on the value of the NO2 observations, which occurs when the observations are made from a different satellite (as in this work) or could potentially occur even with a different instrument on the same spacecraft if the scanning approach differed. Despite these limitations, this is a useful study that contributes to our understanding of combining CO2 and NO2 observations for anthropogenic CO2 source estimation and thus I would recommend it for publication in ACP after some revisions.

Specific Points

Page 2, lines 15-17 is a jumble of references to different techniques and different types of measurements (satellites and airborne). Since the rest of the paper is about satellite observations, the reference to the airborne measurements and associated emission estimates of Krings et al. 2011 and 2018 should either be removed or some explanation is needed as to why they are relevant here. It would also be useful to better distinguish between studies that quantified/estimated emissions versus identification. It might also be helpful to point out the studies that took advantage of atmospheric

imaging capability, which is crucial to the current work.

P2, L27. Specifying that the S5P launch was in October 2017 would help to clarify for the reader why only observations from 2018 were used in this work.

P3, L8: "eight parallelogram-shaped footprints across track with a spatial resolution at ground of ≤1.29 km x 2.25 km"

P3, L13: It would be helpful to add a statement along the lines of "The OCO-2 v9 data set has an improved bias correction approach that results in reduced biases particularly over areas of rough topography."

P4, L2: It would be helpful to provide the SNR or perhaps a relative precision instead of just the random noise, since most CO2 specialists will not have a good grasp of the magnitude with these units.

P4, L14: "50 minutes" According to the figure labels, the time differences range from 6 minutes (Medupi-Matimba) to 35 minutes (Nanjing). Perhaps it would be more informative to state: "each scene observed by OCO-2 is also observed by S5P with a temporal offset ranging from 6 to 35 minutes"?

P5, L5-10: The method described is interesting and very sensible and is one of the strengths of this work.

P5, L29 – P6 L1: The manual adjustment to wind direction but not windspeed is similar to the approach of Nassar et al. (2017) which would be worth acknowledging.

P6, L7: This constant factor of 1.44 taken from Varon et al. (2018) to treat the vertical dimension is a major oversimplification in this work. Varon et al. (2018) simulated CH4 plumes that might be typical of CH4 leaks from infrastructure, thus they deal with smaller spatial scales and little to no temperature contrast. The effective vertical height of emissions will likely be very different when dealing with smokestacks, urban areas or wildfires, as in the present work. In fact, a new paper (Brunner et al. "Accounting for the vertical distribution of emissions in atmospheric CO2 simulations" Atmos. Chem.

Phys., https://doi.org/10.5194/acp-19-4541-2019) that also has links to the Copernicus candidate CO2 Monitoring mission, describes the relevant factors for the vertical distribution of emissions, where different vertical emission profiles for point sources (i.e. a power plant) or area sources (i.e. an urban area) are discussed. The temperature of the emissions and the season are also shown to be important factors. Although detailed study of plume rise is complex and beyond the scope of this paper, and the use of column data reduces the importance of these issues, surely it must be too simple to use a single factor of 1.44 times the 10 m wind speed to represent plume rise from the diversity of source types and geographic locations studied in the present work. According to equation 3, errors in emission estimates will be approximately proportional to the error in wind speed, so getting a realistic wind speed is important.

P7-11, It would be most useful to have the wind direction adjustments clearly stated for every case either in the text or a table.

P8, L7: "larges" -> "largest"

P8, Sec 3.2: The enhancement near Lipetsk is huge and the fit is very good. Are there other sources in addition to the gas-fired power plant and the steel plant that could be relevant, for example, what about the city of Lipetsk (population ~500,000)?

P9, Figure 1: I assume the hashed/shaded region is the Moscow urban area but I am not sure? Can the authors clarify in the figure caption?

P9, Sec 3.4: The OCO-2 flyby of the Matimba and Medupi power plants used in this work is over 80 km away. Nassar et al. (2017) also estimated the emissions from Matimba using OCO-2 data (but version 7) from a direct overpass in 2014 and a close flyby (~7 km away) in 2016. Daily emission estimates from Nassar et al. converted to annual values are ~12 MtCO2/a.

P10, Sec 3.5. The Australian wildfires are clearly an example of an area source not a point source. The NO2 data show structure/heterogeneity in the area source. This

makes it a poor candidate for the modeling approach applied that represents the plume with a Gaussian function. Furthermore, it makes little sense to report emissions in an annual unit in the case of a wildfire, which lasts on the order of days to weeks and would demonstrate temporal variability even over that limited time scale. For fossil fuel $CO_2$ emissions from power plants or cities, there is also periodic (diurnal, weekly and seasonal variability) and non-periodic (plant shutdowns, heating/cooling linked to weather, etc.) variability, which also makes reporting emission rate estimates for shorter time scales more exact from a single overpass.

P14, Figure 6. Nanjing seems to show two maxima in the $NO_2$ image, which is also problematic to represent with a Gaussian function.

P14, L13: Why is 0.5/MtCO2/a the chosen minimum value?

---

## Author Comment (AC1) · 29 May 2019

First of all, we thank reviewer 2 for his/her efforts in carefully reviewing our manuscript and his/her constructive comments.

**Point-by-point answers to the comments of reviewer 2**

**General comments**

**Reviewer 2:** *The number of cases studied is very small. Although 20 promising scenes were identified only six are shown in the manuscript. The authors should include the remaining cases, not as additional examples (except maybe in the supplement), but in order to have more cases to discuss and compare the results.*

**Authors:** Our intention by providing an approximate number of promising cases (20) was to give the reader an impression of how many cases can be expect when applying our current (pre-) selection method to a data set of several months. This does not mean, that we have thoroughly analyzed all of these 20 cases in detail which would make a significant amount of extra work. The focus of our study is on "demonstrating the benefits of simultaneous NO2 and $XCO_2$ measurements rather than on most accurate flux estimates" or on quantifying emissions of an as large as possible number of targets. Therefore, we do not see the immediate necessity to significantly lengthen the paper which also agrees with our interpretation of review 1 stating that "overall, this work is sufficient to demonstrate the value of coincident NO2 and CO2 satellite observations for estimating emissions".

Furthermore, we think that discussing these cases without showing them would be little helpful. As discussed, we use the $NO_2$ measurements "... to i) identify the source of the observed $XCO_2$ enhancements, ii) to exclude interference with potential additional remote upwind sources, iii) to manually adjust the wind direction, and iv) to put a constraint on the shape of the observed $CO_2$ plumes." All of these points, except the last, make use of the shown $NO_2$ maps. This means, when not showing the scenes, the reader would not be able to follow important parts of the analysis.

In case reviewer 2 is curious about !preliminary! results, we have added six more images at the end of this document.

**Reviewer 2:** *The manuscript lacks a detailed comparison of the examples and discussion of the result. A broad summary is given in the conclusions, but this should be moved to a designated section.*

**Authors:** Our manuscript is basically following the IMRaD (Introduction, Methods, Results, and Discussion, *https://en.wikipedia.org/wiki/IMRAD*, Hall, 2012) organizational structure which became the most prominent norm for the structuring of a scientific research article. We only slightly adapted the naming of the section headings of the first hierarchy level: *Methods* became *Data sets and methods* and *Discussion* became *Summary and conclusions.* Renaming the *Method* section is uncritical and many journals prefer heading

style variants like *Methods and materials*, *Materials and methods*, or similar for the *Method* section. Usually, the *Discussion* section includes a summary and the conclusions. Both may or may not be part of the *Discussion* section or separate sections on the same hierarchy level as the *Discussion* section. Therefore, we agree, that the heading *Summary and conclusions* may be misleading and we renamed it to *Discussion and conclusions*. We wanted to keep "Conclusions" within the heading because the ACP template suggests to have a *Conclusions* top-level section.

We do not agree that this section does not include a thorough discussion of the results. Within this section, we compare our results with emission data bases and (in the revised version) also emission estimates of Nassar et al. (2017)): "For Moscow, we derived a cross-sectional flux of $76\pm33\,\mathrm{MtCO_2/a}$ which agrees (within its uncertainty) with ODIAC 2012 emissions of $102\,\mathrm{MtCO_2/a}$ ($88\,\mathrm{MtCO_2/a}$ for 08/2016) but not with EDGAR emissions of $195\,\mathrm{MtCO_2/a}$", "... this estimate agrees with EDGAR emissions of $23\,\mathrm{MtCO_2/a}$ but not with ODIAC emissions of $4\,\mathrm{MtCO_2/a}$", etc. We also discuss scenario specific potential reasons for differences: "... it is interesting to note that Georgoulias et al. (2019) found a strongly increasing trend ... for the tropospheric $NO_2$ concentrations in Baghdad ... hinting at strongly increasing $CO_2$ emissions in Baghdad ...", "... it shall be noted that GFED's emission estimate for the same time interval but one day before the OCO-2 overpass amounts to $252\,\mathrm{MtCO_2/a}$", low wind speed, acute angle of wind direction, etc. Additionally, we discuss the largest contributions to the total uncertainty, the potential effect of co-emitted aerosols, the differences between the emission data bases, the potential difference between cross sectional flux and source emission, etc. This means, whilst the *Results* section basically describes what is shown in the figures, several points within the *Summary and conclusions* section go far beyond a broad summary.

**Reviewer 2:** *Furthermore, the results of the flux estimates without including NO2 fields in the fit should be shown in the results and not only briefly mentioned in the conclusions (P15, L6ff).*
**Authors:** We modified the corresponding paragraph which now reads: "We repeated the flux estimation of all shown scenarios with such a setup and got fluxes of $61\pm27\,\mathrm{MtCO_2/a}$, $63\pm46\,\mathrm{MtCO_2/a}$, $75\pm29\,\mathrm{MtCO_2/a}$, $35\pm9\,\mathrm{MtCO_2/a}$, $166\pm44\,\mathrm{MtCO_2/a}$, and $119\pm28\,\mathrm{MtCO_2/a}$ for the Moscow, Lipetsk, Baghdad, Medupi/Matimba, Australian wildfires, and Nanjing scenario, respectively. The derived fluxes are consistent within their uncertainty with our main results shown in Tab. 1, but the uncertainty contribution due to the noise in the $XCO_2$ data increased by 34% from $4.7\,\mathrm{MtCO_2/a}$ to $6.3\,\mathrm{MtCO_2/a}$ on average."

**Reviewer 2:** *The connection between this study and the proposed CO2M mission, which is emphasized in the abstract and the conclusions, is not well presented. The authors used the NO2 fields to identify the location of the source outside the OCO-2 swath and to screen for potential sources upstream. Both will not be possible with the CO2M mission, if CO2 and NO2 instrument*

*would have the same swath as currently proposed. In addition, it might also not be necessary for CO2M to use NO2 to identify the source outside the swath, because CO2M's swath will be significantly wider than OCO-2's swath. A major advantage of the NO2 observations is likely the potential for improving the fit of the Gaussian (see previous comment), which should be presented more prominently.*

**Authors:** As summarized in the abstract (and the last section), we use the $NO_2$ measurements not only for these two purposes but also to adjust the wind direction and to constrain the shape of the $CO_2$ plumes. CO2M will have a much wider swath compared to OCO-2, so that there is a good chance to see large parts of the plume including the location of the source plus potential upwind sources within the swath even if the $NO_2$ instrument would only cover the same swath as the $CO_2$ instrument. Additionally, it has not yet been finally decided how large the swath width of the $NO_2$ instrument will be, i.e., there is a chance that it might become wider than the $CO_2$ instrument. As discussed within the conclusions, "the noise of the $XCO_2$ retrievals contributes only with $1\,MtCO_2/a$ to $8\,MtCO_2/a$ to the total error". This error enhances by about 34% when ignoring the $NO_2$ measurements. In other words, for the method as presented, other error components are clearly dominating which is why we put the fit improvement not more into focus. However, this will change for CO2M, because "the meteorology related uncertainties will reduce (Varon et al., 2018) because deviations from steady state conditions can average out and are, therefore, less critical if the entire plume structure is sampled rather than only a cross-section". In this case, the uncertainties introduced by the $XCO_2$ retrievals will become more important and the "imaging capabilities (of CO2M) will reduce the uncertainty of the inferred emissions due to measurement noise simply because of the increased number of soundings. Additionally, simultaneous $XCO_2$ and $NO_2$ observations from the same platform will allow stricter constraints on the plume shape."

**Specific comments**

***Reviewer 2:*** *P2, L5ff: Consider re-formulating, e.g.: "... to halve [...] emissions each decade after reaching peak emissions in 2020"*

**Authors:** The sentence now reads: "Actions need to be taken to halve anthropogenic greenhouse gas emissions (including $CO_2$) each decade after reaching peak emissions in 2020 (Rockström et al., 2017)."

***Reviewer 2:*** *P2, L21-23: Consider to remove, because the detailed chemistry seems not relevant for the study.*

**Authors:** We would prefer to keep the paragraph as is, because the complex chemical relations are the reason for potential differences in the plume shape of $CO_2$ and $NO_2$.

**Reviewer 2:**  *P2, L27: Add the resolution of the NO2 instrument.*
**Authors:**  The most important specifications (for our study) of the $NO_2$ instrument are listed in Sec. 2.2 ("... spatial resolution of $3.5\,km \times 7\,km$ at nadir ...").

**Reviewer 2:**  *P2, L29: Clarify the term "localized small scale".*
**Authors:**  The sentence now reads: "... which can be attributed to localized (up to city-scale) emissions ..."

**Reviewer 2:**  *P3, L10f: Consider changing "The data set ..." to "The product ..." to make it clear that the filtering is part of the OCO-2 L2 Lite product.*
**Authors:**  Done.

**Reviewer 2:**  *P4, L1: Please explain the term "viewing angle corrected". Are these geometric air mass factors?*
**Authors:**  In this context, we are interested in the scatter of the slant (not vertical) columns, which is why we corrected only for the viewing angle but not the solar zenith angle, i.e., $NO_2$ slant columns have been corrected for the viewing angle dependent change in the geometric air mass factor. Correcting for changes in the viewing angle is necessary because, otherwise, the inferred scatter would comprise not only changes due to instrumental noise but also the variability of the viewing angle.

**Reviewer 2:**  *P4, L4-9: The paragraph might be easier readable, if it first explains the approach used in this study and briefly contrasts it the "normal" approach.*
**Authors:**  We have the impression that describing first what we have done, renders the description of the "normal" approach a bit superfluous. However, we added "usually" to the first sentence, suggesting that an alternative method will be presented in the following: "In order to extract the tropospheric vertical columns, usually, first the stratospheric contribution ..."

**Reviewer 2:**  *P4, L13: It would be useful to have the time difference between OCO-2 and Tropomi for the six examples presented in the manuscript.*
**Authors:**  The actual time differences per scenario are shown in the figures and the figure captions read: "... and time difference between OCO-2 and S5P overpass are also listed."

**Reviewer 2:**  *P4, L22: Please specify if three times lower resolution is temporal, spatial or both and, if spatial resolution, if it is grid cell area or length. Maybe it's better to just write the resolution of the product.*
**Authors:**  The sentence now reads: "This data archive provides also an uncertainty estimate of the wind information from an ensemble statistic but at a reduced resolution of 0.5°×0.5°×three hours."

**Reviewer 2:** *P5, L7-8: This is not a constraint on the width of the CO2 Gaussian function, if CO2 and NO2 values are fitted simultaneously, but using the same width for fitting both CO2 and NO2 Gaussian function. It would be a constraint if the NO2 plume is fitted first and afterwards the CO2 plume is fitted using the coefficient obtained from the NO2 fit.*

**Authors:** We now decribe more precisely how we constrain the FWHM fit by $NO_2$ only: "We force the FWHM to be constrained entirely by the $NO_2$ measurements by setting the $CO_2$ part of the corresponding Jacobian artificially to zero. However, we expect only little differences to a combined FWHM fit because of the lower relative noise of the $NO_2$ measurements."

**Reviewer 2:** *P5, L13: Please clarify if a Level-2 or Level-3 product is used for the fit.*

**Authors:** We use the co-located $XCO_2$ and $NO_2$ values as described in Sec. 2.3 for the fit. This means the $XCO_2$ values are those given in the OCO-2 L2 product and the $NO_2$ values correspond to averages within the $CO_2$ footprints. Because of OCO-2's much finer spatial resolution, most of the $NO_2$ averages are being build from a single S5P sounding only. The sentence now reads: "Specifically, the co-located $NO_2$ and $XCO_2$ values ..."

**Reviewer 2:** *P5, L22: The equation would be more readable without the unit conversions. The equation could be split in two.*

**Authors:** We removed the unit conversions from Eq. 2 and rephrased the paragraph which now reads:

Integration over the Gaussian enhancement results in the cross-sectional $CO_2$ flux $F_{CO_2}$ (mass of $CO_2$ per time) of the plume depending on the FWHM $a_4$, the amplitude of the $XCO_2$ enhancement $a_7$, the effective wind speed $v_e$ within the plume normal to the OCO-2 orbit, and the number of dry air particles in the atmospheric column $n_e$:

$$ F_{CO_2} = \frac{1}{2} \sqrt{\frac{\pi}{ln(2)}} \frac{M_{CO_2}}{N_A} \, n_e \, a_4 \, a_7 \, v_e $$

Here, $M_{CO_2}$ is the molar mass of $CO_2$ (44.01 g/mol) and $N_A$ the Avogadro constant ($6.02214076 \cdot 10^{13} \, \text{mol}^{-1}$).

...

For a hydrostatic atmosphere with a standard surface pressure of 1013hPa, $n_e$ is about $2.16 \cdot 10^{25} \, \text{cm}^{-2}$ and the cross-sectional $CO_2$ flux $F_{CO_2}$ (Eq. 2) in units of $MtCO_2/a$ becomes approximately

$$ F_{CO_2} \approx 0.53 \, \frac{MtCO_2}{a} \, \frac{a_4}{km} \, \frac{a_7}{ppm} \, \frac{v_e}{m/s} $$

given that the FWHM $a_4$, the amplitude of the $XCO_2$ enhancement $a_7$, and

the effective wind speed $v_e$ are provided in the units km, ppm, and m/s, respectively. As $n_e$ approximately scales with the surface pressure, Eq. 3 may be easily adapted to other meteorological conditions.

**Reviewer 2:**  *P5, L27: Does the OCO-2 product have not air columns that could be used instead?*
**Authors:**  We are using the bias corrected "lite" product which does not include dry air columns but we have to read the ECMWF data in order to obtain the wind information anyhow.

**Reviewer 2:**  *P6, L3: Please add how sensitive the factor (i.e. 0.53) is to surface pressure, to provide a range in which this approximation might be used.*
**Authors:**  We added: "As $n_e$ approximately scales with the surface pressure, Eq. 3 may be easily adapted to other meteorological conditions."

**Reviewer 2:**  *P6, L24: Please state typical values for the uncertainty of the wind speed in the ERA5 data.*
**Authors:**  The sentence now reads: "The uncertainties of the wind components are read from the ECMWF ERA5 data archive resulting in total wind speed uncertainties ranging from $0.18\,\mathrm{m/s}$ to $0.33\,\mathrm{m/s}$ for the analyzed scenarios."

**Reviewer 2:**  *P7, L5: Are the 100 co-locations for Level-2 or Level-3?*
**Authors:**  The $XCO_2$ values are as in the OCO-2 L2 product and the $NO_2$ values correspond to averages within the $CO_2$ footprints (see also discussion above and Sec. 2.3).

**Reviewer 2:**  *P15, L1ff: The major advantage of the NO2 instrument here would be the wider swath and thus having CO2 instrument with a wider swath should bring the same advantage without the need for an additional NO2 instrument. The authors should consider discussing if (or why not) having a CO2 instrument with a wide swath is (not) an option.*
**Authors:**  As visible in the figures and briefly mentioned in Sec. 3.1 ("... larger relative noise of the XCO2 retrievals ..."), the noise of the $NO_2$ retrievals is about 5 times lower than for $XCO_2$ relative to the expected enhancements. This will drastically improve the plume detection and also allow to better constrain the plume shape. Additionally, the background variability due to natural sources/sinks is much larger for $XCO_2$ than for $NO_2$. We modified parts of the abstract, introduction, and summary and conclusions in order to emphasize the difference in the noise performances.

Within the abstract and similarly within the introduction, one now can read: "However, regional column-average enhancements of individual point sources are usually small compared to the background concentration and its natural variability and often not much larger than the satellite's measurement noise. ... It has a short lifetime of the order of hours so that $NO_2$ columns often greatly exceed background and noise levels of modern satellite sensors near sources

which makes it a suitable tracer of recently emitted $CO_2$."

Within the discussion we added: "Despite less strict quality filtering is needed, peak enhancements of $NO_2$ columns near sources can be retrieved from satellites with much lower relative noise than this is the case for $XCO_2$."

***Reviewer 2:*** *P15, L22: Please explain the term "steady state conditions" in the paper.*
**Authors:** The first occurrence of this term is at P6 L12 and we modified it to "... under steady state (temporally invariant) conditions ...".

***Reviewer 2:*** *Figure 1-6: The corrected wind arrow looks very subjectively. It might help to draw the arrow centered on the XCO2 swath where the arrow should be a perpendicular to the plume.*
**Authors:** We do not quite understand the suggestion of the reviewer because non of the arrows should be perpendicular to the plume. For the analyzes of the Gaussian enhancement, we are not considering the across track dimension of the OCO-2 swath. This means, all values are referenced by its distance in flight direction (or time) as plotted in Fig. 1-6c. In the along track dimension, the origin of the arrows is determined by the position of the maximum of the Gaussian $XCO_2$ fit. In the across track dimension a more or less arbitrary sounding is selected.

***Reviewer 2:*** *Table 1: Mention that the single sounding uncertainty (0.4-0.7 ppm) provided the OCO-2 product was used for computing the flux uncertainty, because evaluation with TCCON found a significant larger value (1.3 ppm).*
**Authors:** The table caption now reads: "... the $XCO_2$ precision as reported in the data product, and the $NO_2$ precision as reported in the data product."

***Reviewer 2:*** *Table 1: Add to the columns that estimated cross sections are instantaneous; EDGAR and ODIAC are annual or monthly emissions.*
**Authors:** We added to the table caption: "Note that the cross-sectional flux results correspond to the instantaneous time of the overpass' whilst EDGAR and ODIAC emissions are annual or monthly averages; GFED emissions correspond to six hourly averages (see Sec. 2.4)."

**Technical corrections**

***Reviewer 2:*** *P1, L10: XCO2 was already defined in line 3*
**Authors:** We removed the second occurrence.

***Reviewer 2:*** *P5, L5: Consider changing "... first degree polynomial (i.e. a linear polynomial) ..." to "... a linear polynomial ...".*

**Authors:** Done.

**_Reviewer 2:_** *P5, L6: Consider adding commas before "accounting" and after "values".*
**Authors:** Done.

**References**

Georgoulias, A. K., van der A, R. J., Stammes, P., Boersma, K. F., and Eskes, H. J.: Trends and trend reversal detection in 2 decades of tropospheric $NO_2$ satellite observations, Atmospheric Chemistry and Physics, 19, 6269–6294, https://doi.org/10.5194/acp-19-6269-2019, URL `https://www.atmos-chem-phys.net/19/6269/2019/`, 2019.

Hall, G. M.: How to write a paper (5th ed.), Wiley-Blackwell, BMJ Books, ISBN: 978-0-470-67220-4, 2012.

Nassar, R., Hill, T. G., McLinden, C. A., Wunch, D., Jones, D., and Crisp, D.: Quantifying $CO_2$ emissions from individual power plants from space, Geophysical Research Letters, 44, 2017.

Rockström, J., Gaffney, O., Rogelj, J., Meinshausen, M., Nakicenovic, N., and Schellnhuber, H. J.: A roadmap for rapid decarbonization, Science, 355, 1269–1271, 2017.

Varon, D. J., Jacob, D. J., McKeever, J., Jervis, D., Durak, B. O. A., Xia, Y., and Huang, Y.: Quantifying methane point sources from fine-scale satellite observations of atmospheric methane plumes, Atmospheric Measurement Techniques, 11, 5673–5686, https://doi.org/10.5194/amt-11-5673-2018, URL `https://www.atmos-meas-tech.net/11/5673/2018/`, 2018.

**Figures (!preliminary!)**

[Figure]

Figure 1: As paper Fig. 1 but for Bahrain on January 1, 2018.

[Figure]

Figure 2: As paper Fig. 1 but for Qatar on February 2, 2018.

[Figure]

Figure 3: As paper Fig. 1 but for India on February 25, 2018.

[Figure]

Figure 4: As paper Fig. 1 but for the USA on April 14, 2018.

[Figure]

Figure 5: As paper Fig. 1 but for Australia on April 24, 2018.

[Figure]

Figure 6: As paper Fig. 1 but for Belgium on July 2, 2018.

---

## Author Comment (AC2) · 29 May 2019

First of all, we thank Ray Nassar (reviewer 1) for his efforts in carefully reviewing our manuscript and his constructive comments.

**Point-by-point answers to the comments of reviewer 1**

**Specific points**

**Reviewer 1:** *Page 2, lines 15-17 is a jumble of references to different techniques and different types of measurements (satellites and airborne). Since the rest of the paper is about satellite observations, the reference to the airborne measurements and associated emission estimates of Krings et al. 2011 and 2018 should either be removed or some explanation is needed as to why they are relevant here.*
**Authors:** We cited the publications of Krings et al. (2011, 2018) as examples where assumptions on source position and plume formation have been made. However, as they are not analyzing satellite measurements, we removed them from the revised manuscript.

**Reviewer 1:** *It would also be useful to better distinguish between studies that quantified/estimated emissions versus identification.*
**Authors:** In the revised version, we cite only studies related to quantification/estimation of emissions which is more appropriate in this context: "... and the quantification of anthropogenic emissions a challenging task. Usually, the latter requires knowledge of the source position and assumptions on plume formation (e.g., Nassar et al., 2017; Heymann et al., 2017) or statistical approaches applied on larger areas and/or time periods (e.g., Schneising et al., 2013; Buchwitz et al., 2017)."

**Reviewer 1:** *It might also be helpful to point out the studies that took advantage of atmospheric imaging capability, which is crucial to the current work.*
**Authors:** Here we are aiming to directly lead to emission estimates which make use of simultaneous $NO_2$ measurements. The benefits of imaging capabilities are discussed at p15 l16 of the original manuscript.

**Reviewer 1:** *P2, L27. Specifying that the S5P launch was in October 2017 would help to clarify for the reader why only observations from 2018 were used in this work.*
**Authors:** Done.

**Reviewer 1:** *P3, L8: "eight parallelogram-shaped footprints across track with a spatial resolution at ground of <= 1.29 km x 2.25 km"*
**Authors:** Done.

**Reviewer 1:** *P3, L13: It would be helpful to add a statement along*

*the lines of "The OCO-2 v9 data set has an improved bias correction approach that results in reduced biases particularly over areas of rough topography."*
**Authors:** Done.

*Reviewer 1:* *P4, L2: It would be helpful to provide the SNR or perhaps a relative precision instead of just the random noise, since most CO2 specialists will not have a good grasp of the magnitude with these units.*
**Authors:** We added: "(enhancements near sources often exceed $10^{16}$ molec./cm$^2$)".

*Reviewer 1:* *P4, L14: "50 minutes" According to the figure labels, the time differences range from 6 minutes (Medupi-Matimba) to 35 minutes (Nanjing). Perhaps it would be more informative to state: "each scene observed by OCO-2 is also observed by S5P with a temporal offset ranging from 6 to 35 minutes"?*
**Authors:** About 50 minutes is the maximum co-location time difference for the vast majority of possible OCO-2 observations. 6 to 35 minutes is only the range for the presented cases (which are described later in the paper).

*Reviewer 1:* *P5, L5-10: The method described is interesting and very sensible and is one of the strengths of this work.*
**Authors:** Many thanks.

*Reviewer 1:* *P5, L29 – P6 L1: The manual adjustment to wind direction but not windspeed is similar to the approach of Nassar et al. (2017) which would be worth acknowledging.*
**Authors:** We added: "The manual adjustment to wind direction but not wind speed is similar to the approaches of, e.g., Krings et al. (2011) or Nassar et al. (2017)."

*Reviewer 1:* *P6, L7: This constant factor of 1.44 taken from Varon et al. (2018) to treat the vertical dimension is a major oversimplification in this work. Varon et al. (2018) simulated CH4 plumes that might be typical of CH4 leaks from infrastructure, thus they deal with smaller spatial scales and little to no temperature contrast. The effective vertical height of emissions will likely be very different when dealing with smokestacks, urban areas or wildfires, as in the present work. In fact, a new paper (Brunner et al. "Accounting for the vertical distribution of emissions in atmospheric CO2 simulations" Atmos. Chem. Phys., https://doi.org/10.5194/acp-19-4541-2019) that also has links to the Copernicus candidate CO2 Monitoring mission, describes the relevant factors for the vertical distribution of emissions, where different vertical emission profiles for point sources (i.e. a power plant) or area sources (i.e. an urban area) are discussed. The temperature of the emissions and the season are also shown to be important factors. Although detailed study of plume rise is complex and beyond the scope of this paper, and the use of column data reduces the importance of these issues, surely it must be too simple to use a single factor of*

*1.44 times the 10 m wind speed to represent plume rise from the diversity of source types and geographic locations studied in the present work. According to equation 3, errors in emission estimates will be approximately proportional to the error in wind speed, so getting a realistic wind speed is important.*

**Authors:** As discussed at p6 l9, the focus of our study is "on demonstrating the benefits of simultaneous $NO_2$ and $XCO_2$ measurements rather than on most accurate flux estimates" and we "recognize that uncertainties resulting from our estimate of the effective wind speed's normal may be reduced in the future by improved wind knowledge". At p6 l26, we acknowledge the differences of our study compared to the study of Varon et al. (2018) and account this by enhancing the introduced uncertainty. Within the revised version of the manuscript, we cite Brunner et al. (2019) and the paragraph starting at p6 l5 now reads: "As discussed by Brunner et al. (2019), the plume height (and subsequently the wind speed in plume height) depends on many aspects like emission height, stack geometry, flue gas exit velocity and temperature, meteorological conditions, etc. Some of these parameters are not known for many sources and their explicit consideration would go beyond the scope of this study focusing on demonstrating the benefits of simultaneous $NO_2$ and $XCO_2$ measurements rather than on most accurate flux estimates. Varon et al. (2018) proposed to approximate the effective wind speed within the plume from the 10 m wind by applying a multiplier in the range of 1.3–1.5. Therefore, we decided to use a multiplier of 1.4 for convenience. This empirical relationship accounts, e.g., for plume rise and mixing into altitudes with larger wind speeds. For the present, we consider this approximation adequate for this first study, but we recognize that uncertainties (see next section) resulting from this estimate of the effective wind speed's normal may be reduced in the future by improved wind knowledge."

**Reviewer 1:** *P7-11, It would be most useful to have the wind direction adjustments clearly stated for every case either in the text or a table.*
**Authors:** We added this information for each scenario to the text of the corresponding sections. The adjustments were always between 17° (Baghdad) and 1° (Moscow).

**Reviewer 1:** *P8, L7: "larges" -> "largest"*
**Authors:** Done.

**Reviewer 1:** *P8, Sec 3.2: The enhancement near Lipetsk is huge and the fit is very good. Are there other sources in addition to the gas-fired power plant and the steel plant that could be relevant, for example, what about the city of Lipetsk (population ~500,000)?*
**Authors:** Of course, there are also other $CO_2$ emitting industries in Lipetsk plus traffic etc. However, we do not have access to a emission data base on facility level for Lipetsk. In order to not give the impression that the steel plant and the power plant are the only two emitters, we have rephrased the first sentence of the chapter: "... shows the surrounding of Lipetsk (approx.

0.5 million inhabitants) with, among other industries, the Novolipetsk steel plant and the Lipetskaya TEC-2 gas-fired power plant ...". Additionally, we added the approximate population also to the Moscow, Baghdad, and Nanjing section.

**Reviewer 1:** *P9, Figure 1: I assume the hashed/shaded region is the Moscow urban area but I am not sure? Can the authors clarify in the figure caption?*
**Authors:** We added to the figure caption: "The hatched area corresponds to the urban area (World Urban Areas dataset, Geoportal of the University of California, *https://apps.gis.ucla.edu/geodata/dataset/world_urban_areas*)."

**Reviewer 1:** *P9, Sec 3.4: The OCO-2 flyby of the Matimba and Medupi power plants used in this work is over 80 km away. Nassar et al. (2017) also estimated the emissions from Matimba using OCO-2 data (but version 7) from a direct overpass in 2014 and a close flyby ( 7 km away) in 2016. Daily emission estimates from Nassar et al. converted to annual values are 12 MtCO2/a.*
**Authors:** Our flux estimate corresponds to the combined signal of the Matimba and Medupi power plant. Nevertheless, we added to our discussion: "Nassar et al. (2017) also estimated the emissions from the Matimba power plant (but not Medupi) using OCO-2 $XCO_2$ v7 data. For a direct overpass in 2014 and a close flyby ($\sim 7\,$km away) in 2016 they found fluxes, converted to annual values, of $12.1\pm3.9\,MtCO_2/a$ and $12.3\pm1.2\,MtCO_2/a$, respectively."

**Reviewer 1:** *P10, Sec 3.5. The Australian wildfires are clearly an example of an area source not a point source. The NO2 data show structure/heterogeneity in the area source. This makes it a poor candidate for the modeling approach applied that represents the plume with a Gaussian function. Furthermore, it makes little sense to report emissions in an annual unit in the case of a wildfire, which lasts on the order of days to weeks and would demonstrate temporal variability even over that limited time scale. For fossil fuel CO2 emissions from power plants or cities, there is also periodic (diurnal, weekly and seasonal variability) and non-periodic (plant shutdowns, heating/cooling linked to weather, etc.) variability, which also makes reporting emission rate estimates for shorter time scales more exact from a single overpass.*
**Authors:** We discuss that "the $NO_2$ (and less obvious maybe also the $XCO_2$) cross-section has two maxima" and that "the Gaussian fitting function cannot account for this". However, this is "not reflected in the overall good fit quality ($\chi^2 = 0.6$)" for the $XCO_2$ fit. This means, within the noise of the $XCO_2$ data, we cannot expect to significantly improve the $XCO_2$ fit by a more complex plume model. We rephrased the corresponding section which now reads: "The $NO_2$ (and less obvious also the $XCO_2$) cross-section has two maxima which cannot be accounted for by the Gaussian fitting function. However, this is not reflected in the good $XCO_2$ fit quality ($\chi^2 = 0.6$), but should be taken into account when valuing the results." The unit $MtCO_2/a$ is not unusual especially

for fluxes of power plants or other anthropogenic $CO_2$ emitters. Likewise driving a car at 50miles/hour does not mean that you actually drive the car one hour at that speed, our flux estimates are snapshots for the time of the overpass and are not necessarily representative for the annual average. This is particularly true for a wildfire but of course also for, e.g., a power plant if emissions vary in time. Within the section summary and conclusions we state that "our estimates are valid only for the time of the overpass...". Additionally, we emphasize this point within the sections for the Australian wildfires ("for the snapshot of the overpass, we computed a cross-sectional $CO_2$ flux ..."). The revised version makes this point also clear in a modified caption of Tab. 1: "Note that the cross-sectional flux results correspond to the instantaneous time of the overpass' whilst EDGAR and ODIAC emissions are annual or monthly averages..."

*Reviewer 1:* *P14, Figure 6. Nanjing seems to show two maxima in the NO2 image, which is also problematic to represent with a Gaussian function.*
**Authors:** As we do not fit the entire plume of $CO_2$ or $NO_2$ with a Gaussian plume model, additional upwind maxima in the $NO_2$ image pose no principle problem for the analysis of the plumes cross sectional flux. If an additional maximum is coming from an additional source, the source attribution becomes difficult. If it results from non steady state meteorological conditions (e.g., accumulated $NO_2$ during calm conditions), the plumes cross sectional flux may still be correctly derived but may become a poor estimate for the actual emission. For the Nanjing scene, EDGAR and ODIAC emissions suggest multiple significant emitters and we discuss in the section summary and conclusions: "... the scene includes a larger area of overlaying sources, making source attribution difficult."

*Reviewer 1:* *P14, L13: Why is 0.5/MtCO2/a the chosen minimum value?*
**Authors:** For most meteorological conditions, a source of $0.5\,\mathrm{MtCO2/a}$ per grid box is typically well below the detection limit of OCO-2. Additionally, given a limit of $0.5\,\mathrm{MtCO2/a}$, the shown maps include a reasonable small amount of non-empty grid boxes so that the reader is easily able to find the largest emitters.

**References**

Brunner, D., Kuhlmann, G., Marshall, J., Clément, V., Fuhrer, O., Broquet, G., Löscher, A., and Meijer, Y.: Accounting for the vertical distribution of emissions in atmospheric $CO_2$ simulations, Atmospheric Chemistry and Physics, 19, 4541–4559, 2019.

Buchwitz, M., Schneising, O., Reuter, M., Heymann, J., Krautwurst, S., Bovensmann, H., Burrows, J. P., Boesch, H., Parker, R. J., Somkuti, P., Detmers, R. G., Hasekamp, O. P., Aben, I., Butz, A., Frankenberg, C., and Turner, A. J.: Satellite-derived methane hotspot emission estimates using a fast data-driven method, Atmospheric Chemistry and Physics, 17, 5751–5774, https://doi.org/10.5194/acp-17-5751-2017, URL `https://www.atmos-chem-phys.net/17/5751/2017/`, 2017.

Heymann, J., Reuter, M., Buchwitz, M., Schneising, O., Bovensmann, H., Burrows, J., Massart, S., Kaiser, J., and Crisp, D.: $CO_2$ emission of Indonesian fires in 2015 estimated from satellite-derived atmospheric $CO_2$ concentrations, Geophysical Research Letters, 44, 1537–1544, 2017.

Krings, T., Gerilowski, K., Buchwitz, M., Reuter, M., Tretner, A., Erzinger, J., Heinze, D., Pflüger, U., Burrows, J. P., and Bovensmann, H.: MAMAP - a new spectrometer system for column-averaged methane and carbon dioxide observations from aircraft: retrieval algorithm and first inversions for point source emission rates, Atmospheric Measurement Techniques, 4, 1735–1758, https://doi.org/10.5194/amt-4-1735-2011, URL `https://www.atmos-meas-tech.net/4/1735/2011/`, 2011.

Krings, T., Neininger, B., Gerilowski, K., Krautwurst, S., Buchwitz, M., Burrows, J. P., Lindemann, C., Ruhtz, T., Schüttemeyer, D., and Bovensmann, H.: Airborne remote sensing and in situ measurements of atmospheric $CO_2$ to quantify point source emissions, Atmospheric Measurement Techniques, 11, 721–739, https://doi.org/10.5194/amt-11-721-2018, URL `https://www.atmos-meas-tech.net/11/721/2018/`, 2018.

Nassar, R., Hill, T. G., McLinden, C. A., Wunch, D., Jones, D., and Crisp, D.: Quantifying $CO_2$ emissions from individual power plants from space, Geophysical Research Letters, 44, 2017.

Schneising, O., Heymann, J., Buchwitz, M., Reuter, M., Bovensmann, H., and Burrows, J. P.: Anthropogenic carbon dioxide source areas observed from space: assessment of regional enhancements and trends, Atmospheric Chemistry and Physics, 13, 2445–2454, https://doi.org/10.5194/acp-13-2445-2013, URL `http://www.atmos-chem-phys.net/13/2445/2013/`, 2013.

Varon, D. J., Jacob, D. J., McKeever, J., Jervis, D., Durak, B. O. A., Xia, Y., and Huang, Y.: Quantifying methane point sources from fine-scale satellite observations of atmospheric methane plumes, Atmospheric Measurement Techniques, 11, 5673–5686, https://doi.org/10.5194/amt-11-5673-2018, URL `https://www.atmos-meas-tech.net/11/5673/2018/`, 2018.

---

## Author Response (AR2)

Many thanks again to reviewer 2 for his/her efforts and his/her constructive comments.

**Point-by-point answers to the comments of reviewer 2**

**Minor corrections**

***Reviewer 2:*** *P1, L22ff: The last two sentences are somewhat disconnected from the previous sentences. My suggestion would be to modify the sentence as follows: The flux uncertainties are expected to be reduced by the planned European Copernicus anthropogenic CO2 monitoring mission (CO2M), which will not only [...]*
**Authors:** Perfect, included as suggested.

***Reviewer 2:*** *P4, L6ff: Please add an explanation of "viewing angle corrected" to the manuscript. For example: "The random noise of our S5P slant columns was estimated from the scatter of observations over a clean Pacific region (...). To account for the viewing angle dependency of the slant columns a geometric air mass factor was computed using only the instrument viewing. The evaluation suggests that the random noise () of our S5P slant columns is typically $5 \times 10^{14}$ molec./cm2 while enhancements near sources often exceeds $1 \times 10^{16}$ molec./cm2."*
**Authors:** Implemented basically as suggested.

***Reviewer 2:*** *P6, L6: The exponent of Avogadro's number should be 23 instead of 13.*
**Authors:** Done. Thanks for spotting this typo.

***Reviewer 2:*** *P8, L7: Please consider re-formulating the following sentence to make your intention clearer as written in your response. For example: "In total, we manually identified about 20 promising scenarios in the time period from 01/2018 to 08/2018 of which six example were selected and analyzed for this study."*
**Authors:** Done, the sentence now reads: "In total, we manually identified about 20 promising scenarios in the time period 01/2018 to 08/2018 of which we selected and analyzed six examples for this study."

Many thanks again to reviewer 2 for his/her efforts and his/her constructive comments.

**Point-by-point answers to the comments of reviewer 2**

**Minor corrections**

**Reviewer 2:** *P1, L22ff: The last two sentences are somewhat disconnected from the previous sentences. My suggestion would be to modify the sentence as follows: The flux uncertainties are expected to be reduced by the planned European Copernicus anthropogenic CO2 monitoring mission (CO2M), which will not only [...]*
**Authors:** Perfect, included as suggested.

**Reviewer 2:** *P4, L6ff: Please add an explanation of "viewing angle corrected" to the manuscript. For example: "The random noise of our S5P slant columns was estimated from the scatter of observations over a clean Pacific region (...). To account for the viewing angle dependency of the slant columns a geometric air mass factor was computed using only the instrument viewing. The evaluation suggests that the random noise () of our S5P slant columns is typically 5×1014 molec./cm2 while enhancements near sources often exceeds 1×1016 molec./cm2."*
**Authors:** Implemented basically as suggested.

**Reviewer 2:** *P6, L6: The exponent of Avogadro's number should be 23 instead of 13.*
**Authors:** Done. Thanks for spotting this typo.

**Reviewer 2:** *P8, L7: Please consider re-formulating the following sentence to make your intention clearer as written in your response. For example: "In total, we manually identified about 20 promising scenarios in the time period from 01/2018 to 08/2018 of which six example were selected and analyzed for this study."*

[revised manuscript text omitted]